# Atmospheric Conditions Leading to an Exceptional Fatal Flash Flood in the Negev Desert, Israel

Uri Dayan[1], Itamar M. Lensky[2], Baruch Ziv[3], Pavel Khain[4]

[1]Department of Geography, The Hebrew University of Jerusalem, Jerusalem, 9070227, Israel
[2]Department of Geography and Environment, Bar-Ilan University, Ramat-Gan, 5290002, Israel
[3]Department of Natural Sciences, The Open University of Israel, Raanana, Israel
[4]Israel Meteorological Service, Israel

*Correspondence to*: Itamar M. Lensky (itamar.lensky@biu.ac.il)

**Abstract.** The study deals with an intense rainstorm that hit the Middle East between 24 and 27 April 2018 and took the lives of 13 people, 10 of them in 26 April during the deadliest flash flood in Tzafit Basin (31.0°N 35.3°E), the Negev Desert. The rainfall observed in the southern Negev was comparable to the long-term annual rainfall there, with intensities exceeding 75 years return period. The timing of the storm, at the end of the rainy season, when rain is relatively rare and spotty, raises the question what were the atmospheric conditions that made this rainstorm one of the most severe late spring storms.

The synoptic background was an upper-level cutoff low that formed south of a blocking high, which developed over Eastern Europe. The cut-off low entered the Levant near 30ºN latitude, and slowed its movement from ~10 to <5 ms$^{-1}$, and so extended the duration of the storm over the region. The dynamic potential of the cutoff low, as estimated by its curvature vorticity, was the largest among the 12 late spring rainstorms that occurred during the latest 33 years. The lower levels were dominated by a cyclone centred over north western Saudi Arabia, producing north-westerly winds that advected moist air from the Mediterranean inland. During the approach of the storm, the atmosphere over Israel became unstable, with instability indices reaching values favourable for thunderstorms (e.g., CAPE > 1500 J Kg$^{-1}$, LI = 4 K) and the precipitable water reaching 30 mm. The latter is explained by lower-level moisture advection from the Mediterranean and an additional contribution of mid-level moist air transport entering the region from the east. Three major rain centres were active over Israel during April 26, only one of them was orographic and the other two were triggered by instability and meso-scale cyclonic centers. The build-up of the instability is explained by a negative upper-level temperature anomaly over the region, caused by a northerly flow east of a blocking high that dominated Eastern Europe, and ground warming during several hours under clear sky. The intensity of this storm is attributed to an amplification of a mid-latitude disturbance, which produced a cutoff low, with its implied high relative vorticity, low upper-level temperatures and slow progression. All these,

combined with the contribution of moisture supply, led to intense moist convection that prevailed over the region three days successively.

## 1 Introduction

On April 26, 2018 an extremely intense rainstorm hit Israel. Thirteen people were killed in this storm, ten of them by a raging flash flood in the Tzafit Basin (31.0°N 35.3°E), northeast of the Negev Desert (denoted in Fig. 1). The rainfall depth during this storm exceeded 20 mm over above most parts of the country except the north-western Negev and the southern tip of the Negev, over most of which the monthly long-term mean does not exceed 10 mm (Porat et al., 2018).

Flash flood forecast and warning is a challenging task, in particular over remote arid areas, where meteorological radar coverage is sparse. Though flash flood warning has much improved in the last few decades, it is still the most weather-related fatal hazard globally (Montz et al. 2002), with 1,075 fatalities between 1996 and 2014 in the United States alone (Terti et al. 2019). A significant part of flash flood victims is caused by incorrect assessment or misjudging of the risks of these rapidly evolving phenomena (Sharif et al. 2015; Becker et al. 2015; Diakakis et al. 2018).

The study focuses on the Negev Desert and the Judean Desert (Fig. 1), hereafter referred to as the 'study region'. The climatic regimes of the study region span from semiarid in the north to arid in the centre and the south. The majority of the annual precipitation in Israel is associated with Mediterranean cyclones, while reaching its eastern part (i.e., Cyprus Lows, HMSO 1962; Saaroni et al. 2010; Zappa et al. 2015). Two-thirds of the rainfall occur during December through February (Alpert et al. 2004; Ziv et al. 2006). During the late spring (Apr-May), the rainfall over the northern and central parts of the Negev Desert constitutes 4 - 9% of the annual average (5 - 10 mm). In spite of these negligible rain amounts, the number of flash flood events cannot be ignored.

The flood regime in the study region was analysed by Kahana (1999), based on 37 hydrometric stations operated by the Israeli Hydrological Service. He identified 59 "major floods" in which the recorded peak discharge reached the magnitude of a 5-year recurrence interval (for the period 1947- 1994) at least in one watershed. 14% of these major floods occurred during the late spring.

The main source of major floods in the late spring over the study region is the active "Red Sea Trough" (ARST, Kahana et al., 2002). The RST is a low-pressure system extending from south toward the Eastern Mediterranean (EM) and the Levant (Ashbel 1938; Kahana et al. 2002; Tsvieli and Zangvil 2005). The ARST is most frequent during fall and spring (Sharon, 1978; Sharon and Kutiel, 1986; Dayan et al., 2001). The ARST is accompanied by

a pronounced upper-level trough that develops over Egypt, and at times, initiates convective storms over the Levant. The secondary source of major floods is the Syrian Low - a derivative of the Cyprus Low. Syrian lows are Mediterranean mid-latitude cyclones that approach Syria and deepen, in contrast to the normal conditions, in which Mediterranean cyclones tend to decay upon reaching Syria (Kahana et al, 2002). The implied winds over Israel during Syrian Lows are north-westerly, enriched with moisture originating from the Mediterranean that cross the Judean and the Negev Mountains perpendicular to the terrain upslope, supporting orographic rains.

The storm analysed here was associated with a Syrian low, accompanied by an upper-level closed cyclone, resembling the features of a 'cutoff low' (as is shown in Sec. 3 below). A cutoff low is defined as a closed cyclone in the upper levels, overlapping with a PV maximum (e.g., Hoskins et al., 1985). Cutoff lows are considered as favourable for severe weather over the Mediterranean Basin (Porcu et al., 2007). For instance, in south Portugal cutoff lows were found to be a major source for heavy rainfall (Fragoso and Tildes Gomes, 2008). Porcu et al. (2007) identified 273 cutoff lows over the entire Mediterranean Basin during the period 1992–2001. As for the Levant (30-35°N, 30-35°E), they found 6 cases (average of 0.6 $y^{-1}$) during the months April-September. Since the Levant is free of significant upper-level cyclonic system (and also of rain) during the summer months June-September (Kushnir et al., 2017), this result reflects, actually, the events of April – May.

This storm was severe in several aspects. One is the number of fatalities, 13, which is record breaking for Israel. Second, the north eastern part of the Negev Desert experienced rain intensities reaching 75-100-year return period, resulting in discharge magnitudes of 10-50-year return period (Rinat et al., 2020). The third aspect is the rainfall totals for the storm, which reached 40-50 mm over wide parts of the study region, i.e., 10 times the monthly long-term mean over the study region (Porat et al., 2018).

The aim of this study is to analyse the storm in its seasonal context and to identify the atmospheric conditions that explain its severity. The article is organized as follows: in the next section, we present the data and methods used to analyse the event. Section 3 describes the event through observational evidence. In section 4 we discuss the unique conditions that lead to the severe convection, and the sources of moisture for the rain formation in this storm. The main results are summarised in Sec. 5.

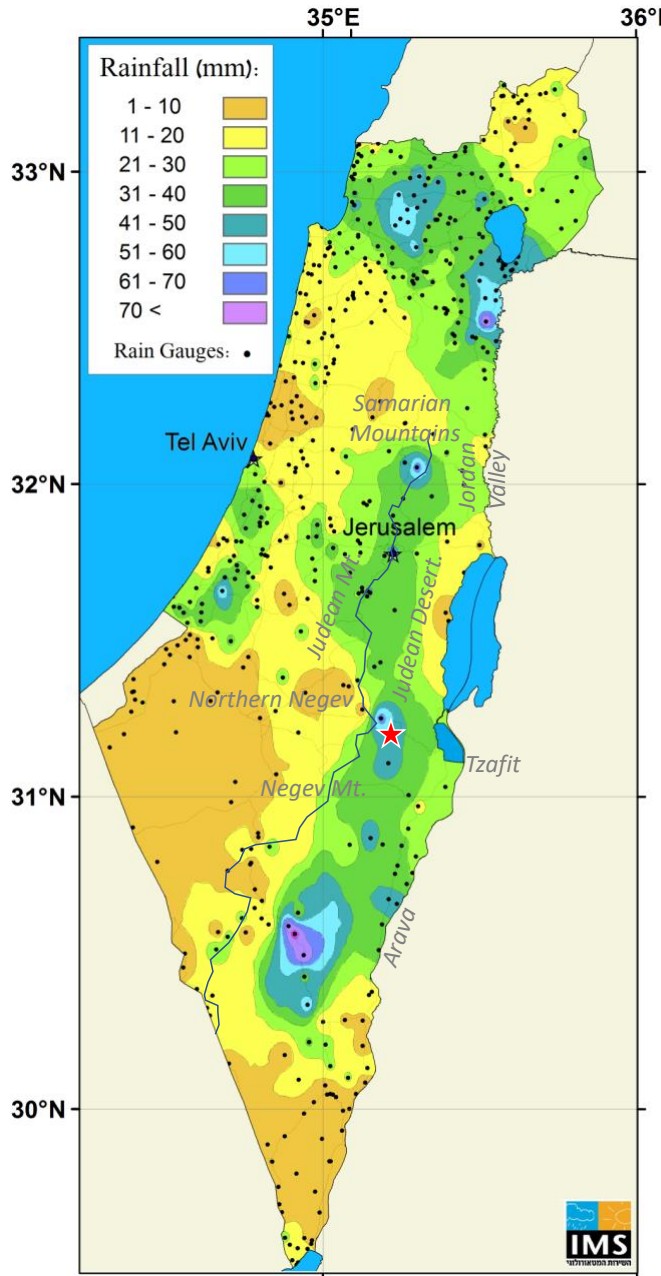

**Figure 1: Rainfall distribution over Israel for the period of April 25 06 UTC to April 27 06 UTC 2018, from the IMS radar and rain gauge measurements based on the INCA system integration method (Haiden et al. 2011). The black line indicates the water divide of the Samarian, Judean and the Negev Mountains.**

## 2 Data and Methods

The study area is Israel, with an emphasis given to the Negev Desert and the Judean Desert (Fig. 1). The study period, April 24-27, 2018, corresponds to a storm that influenced the EM, in particular April 26, when the Tzafit flood took place. This storm is one of 12 storms spread over 27 days ('reference storms', hereafter) with the highest surface runoff observed over the study area in the late spring during 1986 – 2018, based on the hydrometric stations of the Israeli Hydrological Service.

The rainfall data is based on the Israeli Meteorological Service (IMS) rain measuring network and radar data (Porat et al., 2018). The spatial resolution of the radar is 250 m in the radial direction and 1° in the azimuthal. This data is rectified to 1×1 km grid, and its temporal resolution is 5 min. The rain intensity was derived from the radar data (Marra and Morin, 2015).

The atmospheric processes responsible for flash flood-producing rainstorms combine synoptic and sub-synoptic scales (Bardossy and Filiz, 2005). This also holds specifically for severe convection during the spring over the Negev Desert (Dayan and Morin, 2006; Armon et al., 2018). The data for the synoptic maps, including sea level pressure (SLP) and upper-level fields and geopotential height (gph) were taken from NCEP/NCAR reanalysis 1 database, at 2.5°×2.5° spatial resolution (Kalnay et al. 1996; Kistler et al. 2001). The data for the mesoscale maps including wind, relative and potential vorticity, temperature and relative humidity, are taken from ERA5 ECMWF reanalysis data base (C3S, 2017, https://cds.climate.copernicus.eu/cdsapp#!/dataset/reanalysis-era5-pressure-levels) at 30×30 km resolution. SLP maps from the two data bases for the pertinent region and period were compared for consistency and were found reasonably fit (not shown).

For finer resolution analysis we used the COSMO regional model (www.cosmo-model.org), operational at the IMS, covering the Eastern Mediterranean domain (25-39E/26-36N, Khain et al., 2020b) with 2.8×2.8 km resolution. The model analyses are created by using a continuous assimilation of the radar and rain gauges composite via latent heat nudging (Stephan et al., 2008, Khain et al., 2020a). The boundary and initial conditions are taken from the European Centre for Medium-Range Weather Forecasts (ECMWF) Integrated Forecasting System (IFS) (www.ecmwf.int/en/forecasts/documentation-and-support). The COSMO model is based on the primitive thermo-hydrodynamic equations that describe non-hydrostatic compressible flow in a moist atmosphere. Its vertical extension reaches 23.5 km (~30 hPa) with 60 model levels, including 12 levels between the surface and 900 hPa and 15 levels between 900 and 500 hPa, being able to capture the PBL effects (Uzan et al., 2020).

According to the quasi-geostrophic theory, the relative vorticity is a general measure for cyclone intensity and activity (e. g., Holton 1992). The curvature term dominates near cyclone centres and within meandering flow, whereas the shear term dominates near jets (Uccellini and Kocin, 1987). In the cases analysed here, an upper-level cyclone was found over the Levant or its proximity, accompanied by a southward shift and a weakening of the subtropical jet, and therefore, the curvature vorticity was considered as dominant.

Following the above, the vorticity here is approximated by a 'measure of curvature vorticity' (MCV, hereafter), which is calculated on the periphery of the upper-level (500 hPa) cyclones, based on the relation

$$\xi_c = V/R \;, \tag{1}$$

where $\xi_c$ is the curvature vorticity, $V$ is the tangential wind speed, and $R$ is the cyclone radius, measured on the outer-most closed isohypse (when derived with 15 m intervals). The tangential wind speed was calculated using the geostrophic relation, i.e.,

$$V \cong g \cdot D/(R \cdot f) \;, \tag{2}$$

where g and $f$ are the gravitational acceleration and Coriolis parameter, respectively, and $D$ is the depth of the cyclone, determined by the difference between its central height and that of the outer-most closed isohypse. Inserting $V$ from Eq. (2) in Eq. (1) yields:

$$MCV = g \cdot D/(R^2 \cdot f) \quad. \tag{3}$$

The stability indices were derived from the sounding data of Beit Dagan station, Israel (32.5°N, 34.8°E), retrieved from the Department of Atmospheric Sciences, The University of Wyoming, at: http://weather.uwyo.edu/upperair/sounding.html. These are:

• Lifted Index (LI, Galway 1956): Temperature difference between the environment and an air parcel lifted adiabatically from 2 m above the surface to 500 hPa. Negative values indicate instability.

• Showalter stability index (SI, Showalter 1953): Similar to the lifted index (LI), but using a parcel lifted from 850 hPa to 500 hPa. As for the LI, negative SI values indicate instability.

• Convective Available Potential Energy (CAPE, Moncrieff and Miller 1976): The integrated energy excess of an air parcel lifted adiabatically with respect to its environmental temperature profile. Values in the order of hundreds of J Kg$^{-1}$ and more correspond to severe weather.

Maps of the Modified K-Index (MKI) were derived from the ERA5 ECMWF data that cover the Earth on a 30km grid. The original K-Index (KI; Geer 1996), which combines instability and moisture availability, is used to predict severe thunderstorms in the US. The modified version of Haratz et al. (2010), MKI, is the version adapted for the eastern Mediterranean:

$$MKI = (T_{500} - T_{850}) \cdot RH_{850,700} + Td_{850} - (T_{700} - Td_{700}) , \qquad (4)$$

where $T$ and $Td$ are temperature and dew point, respectively, $RH$ is relative humidity and the subscripts refer to the respective pressure level (hPa). The modification gives more weight to the relative humidity at the 850 and 700 hPa levels.

In addition, we also used Precipitable Water (PW, Liu, 1986): The depth of water in the atmospheric column, if all the water in that column were precipitated as rain. The values were calculated from the soundings of Beit Dagan. The METEOSAT Second Generation Water vapour imageries were retrieved from EUMETSAT data centre (https://navigator.eumetsat.int/product/EO:EUM:DAT:MSG:HRSEVIRI) and were used to represent main cloud patterns and mid-level moisture. The IR channel was converted to brightness temperatures (Lensky and Rosenfeld, 2008) and used for estimating the height of the cloud tops.

Air back-trajectories for detecting moisture transport were retrieved from the site of NOAA HYSPLIT MODEL (Stein et al., 2015, https://www.ready.noaa.gov/HYSPLIT.php) Global Data Assimilation System (GDAS 1°×1° resolution, on pressure levels), which is used by the National Center for Environmental Prediction (NCEP) Global Forecast System (GFS) model to place observations into a gridded model space for initializing weather forecasts with observed data. The 'Model vertical velocity' (derived from the gridded data) option was used.

## 3 Observational analysis

### 3.1 Synoptic evolution

During April 25–27 2018 the Levant was dominated by an upper-level closed cyclone, accompanied by a lower-level cyclone, located east of Israel (see Fig. 2).

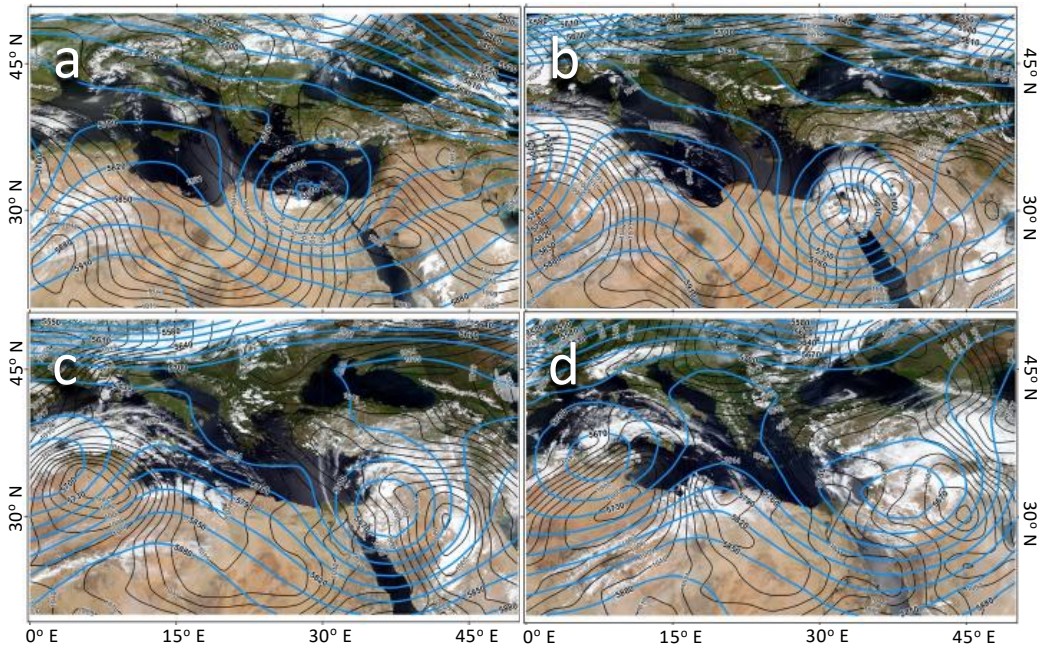

**Figure 2: Geopotential height (GPH) (blue lines) at 500 hPa (m) and SLP (hPa) (black lines) projected on MODIS imagery, in 24 h intervals, for April 24–27 2018, 12UTC (a–d, respectively).**

Prior to the storm, on April 24, 2018, the lower levels were influenced by a RST. The axis of the RST was located

east of the Levant, enclosing a shallow cyclone, with a central pressure of 1008 hPa, over north western Saudi Arabia (see Fig. 2a), transporting warm and dry air from the east toward the Levant. At the 500 hPa level, a closed cyclone was approaching the region from the central Mediterranean. During April 25 the upper-level cyclone slowed its progress, from ~ 9 ms$^{-1}$ to ~ 4 ms$^{-1}$, while arriving to the Egyptian coast (see Fig. 3). The surface shallow cyclone deepened, attaining a central pressure of 1000 hPa, and moved northward, to Iraq, resembling the Syrian

Low (Kahana et al. 2002). The winds over the south-eastern Mediterranean backed to north-westerly, hence advecting moist air from the Mediterranean inland, to central and south Israel. The rain in Israel started at April 25 and stopped at April 27 (Porat et al., 2018). The spatial distribution of the rain during the entire storm is shown in Fig. 1. The maximum rainfall was obtained east of the water divide of the Negev Mountains, and the second most intense one was found in the Jordan valley, 200 m BSL.

During the following day, April 26, the upper-level cyclone continued its slow eastward progression (Figs. 2c and 3). The weakening of the horizontal pressure gradient at the surface in April 27 (compare Figs. 2c and 2d), in

tandem with a flattening of the 500-hPa cyclone and its movement further eastward, led to the cessation of the stormy weather over Israel.

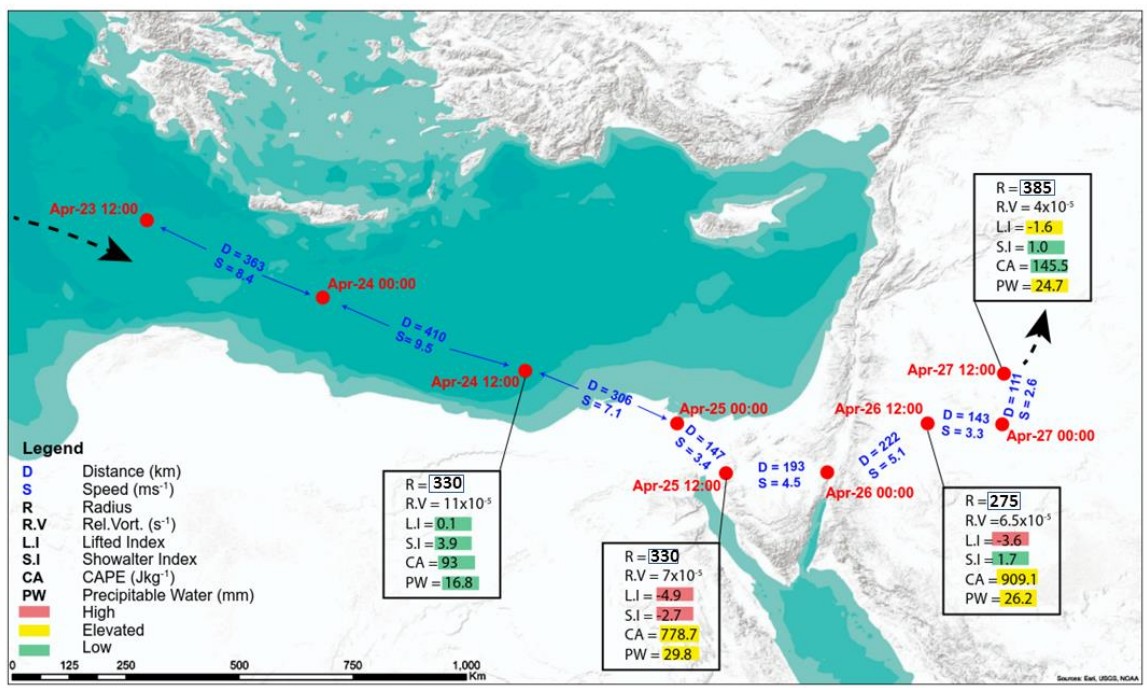

**Figure 3: Track of the upper-level cyclone (500 hPa GPH) during 24–27 April 2018, in 12 hours intervals. The instantaneous speed (ms⁻¹) and distance (km) spanned during each increment are denoted by S and D, respectively. For each 24 hours increment, the radius of the upper-level low (km), the maximum relative vorticity (s⁻¹), the precipitable water (PW, in mm) and 3 thermodynamic indices (LI, SI, and CA) as**
**calculated from the sounding of Beit Dagan are specified. "High", "Elevated" and "Low" values of the indices are highlighted in red, yellow and green, respectively.**

The upper-level cyclone, beside its geopotential depth (up to 100 m, Fig. 2), was accompanied by an isolated PV maximum, as seen in the 300 hPa map for April 26 00 UTC (Fig. 4), indicating its being a cutoff low (Hoskins et
al. 1985). Moreover, its evolution took place southeast of an upper-level blocking high that covered East Europe during April 24 and 25, as can be inferred from the PV distribution seen in Figs. 5a-c. The interaction between the blocking high and the cutoff low and its contribution to the slowing of the cutoff low while approaching the Levant are elaborated in Sec. 4 below.

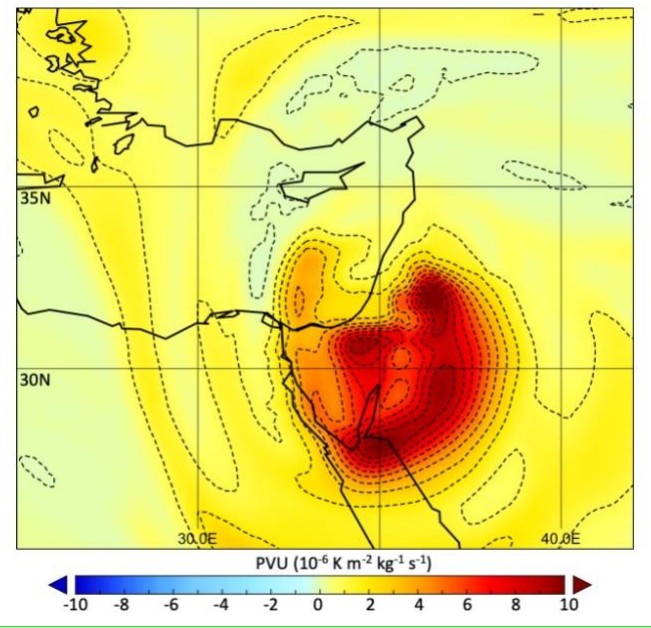

**Figure 4: Potential Vorticity (PVU) at 300 hPa for April 26 00 UTC, taken from ERA-5, at 30 km spatial resolution. The major feature is a closed positive anomaly (with maximum of K m$^{-2}$ Kg$^{-1}$ s$^{-1}$), corresponding to a cutoff low, centred over north-western Saudi Arabia.**

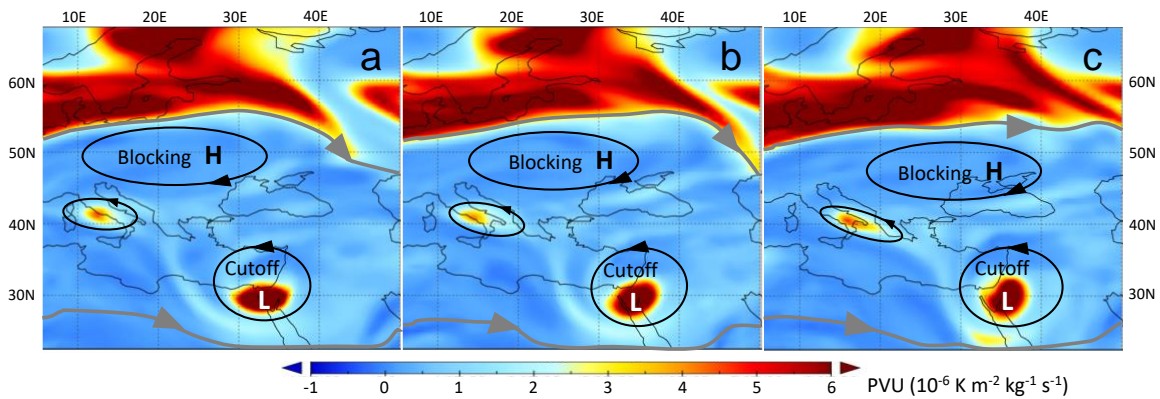

**Fig. 5: PV distribution at 300 hPa, starting from April 25 2018, 12 UTC, in 6 hour intervals, covering Europe and the Mediterranean Basin, on which a conceptual sketch of a dipole type block (following Yamazaki and Itoh, 2013) is superposed. A cutoff low is seen over the south-eastern Mediterranean and a blocking high over Eastern Europe, forming a dipole. The arrows show the induced flow of each of the vortices.**

The synoptic-climatologic background for this storm is compared with the other rain-storms, comprising the set of 12 'reference storms', all of them occurred in the late spring and produced considerable surface runoff over the

study area (see Sec. 2 above). In all of the reference storms, except one, upper-level closed cyclones were found over the EM or the Levant. The cyclones' depth varied between 10 and 130 m, with an average of 52 m, and their radii ranged between 200 and 900 km, with an average of 600 km (Table 1). The MCV for the various days belonging to the reference storms varied between 0.8 and $3.8\times10^5$ s$^{-1}$, with an average of $2.3\times10^5$ s$^{-1}$. The maximum value was observed during the present storm. The MCV for the present storm was ~1/2 that of the respective maximum relative vorticity (Fig. 3), presumably due to the spatial smoothing implied by the derivation method of the MCV.

During the approach of the cutoff low, the atmosphere over the study region became more and more unstable. Figure 3 shows the values of leading instability indices (specified in Sec. 2) derived from Beit Dagan soundings, in 24-hour increments. They reflect a gradual increase toward April 25 12 UTC, while the upper-level cyclone approached the Sinai Peninsula. At that time, the value of CAPE was 779 J Kg$^{-1}$, of LI: -4.9 K and of SI: -2.7 K, indicating a potential for thunderstorms. In April 26, the CAPE rose to 909 J Kg$^{-1}$.

The transition from the dry conditions that persisted at April 24 to the wet conditions that developed toward April 25 is manifested by a sharp increase in the precipitable water (PW) in Beit Dagan, from 17 mm (April 24 12 UTC) to 30 mm in April 25 12 UTC (27 mm in April 26). The moisture increased, presumably due to the north-westerly wind, as evidenced by the air trajectories entering the study area at the heights of 600 and 1000 m. The trajectory that entered the region at 1400 m originated from Iraq, entered northern Israel from the east and, after turning cyclonically, penetrated the Negev Desert from northwest (Fig. 6). The signature of the segment of the latter trajectory, extending from Iraq toward north Israel, is clearly seen as a bright band in the satellite imagery of the water vapour channel for April 25 21 UTC (12 hours prior to Tzafit flood, Fig. 7), reflecting a moisture strip.

**Table 1.** Depth, radius and the measure of curvature vorticity (MCV) of the 500 hPa cyclones for the 27 days, in which the highest surface runoff was observed over the study area in the late spring during the period 1986–2018 (the reference storms). Note that in 24 of the days an upper-level cyclone was identified.

| Storm No. | Date | GPH Depth (m) | Radius (km) | MCV ($\times10^5$ s$^{-1}$) |
|---|---|---|---|---|
| 1 | 19860402 | 70 | 650 | 2.3 |
| 1 | 19860403 | 25 | 400 | 2.1 |
| 2 | 19860408 | 45 | 650 | 1.4 |
| 2 | 19860409 | 45 | 900 | 0.8 |
| 3 | 19970514 | no | no | |
| 3 | 19970515 | no | no | |
| 4 | 20010404 | 130 | 900 | 2.2 |
| 5 | 20010430 | 15 | 250 | 3.3 |
| 5 | 20010501 | 25 | 350 | 2.8 |

| 5  | 20010502 | 90  | 850 | 1.7 |
|----|----------|-----|-----|-----|
| 5  | 20010503 | 90  | 750 | 2.2 |
| 6  | 20070512 | 50  | 580 | 2.0 |
| 7  | 20110404 | 70  | 900 | 1.2 |
| 7  | 20110405 | 20  | 280 | 3.5 |
| 8  | 20140507 | 15  | 330 | 1.9 |
| 8  | 20140508 | 50  | 550 | 2.2 |
| 9  | 20140511 | 35  | 500 | 1.9 |
| 9  | 20140512 | 10  | 200 | 3.4 |
| 10 | 20150416 | 30  | 350 | 3.3 |
| 11 | 20160410 | no  | no  |     |
| 11 | 20160411 | 45  | 550 | 2.0 |
| 11 | 20160412 | 50  | 650 | 1.6 |
| 11 | 20160413 | 40  | 550 | 1.8 |
| 11 | 20160414 | 45  | 520 | 2.3 |
| 12 | 20180425 | 100 | 600 | 3.8 |
| 12 | 20180426 | 80  | 550 | 3.6 |
| 12 | 20180427 | 70  | 600 | 2.6 |

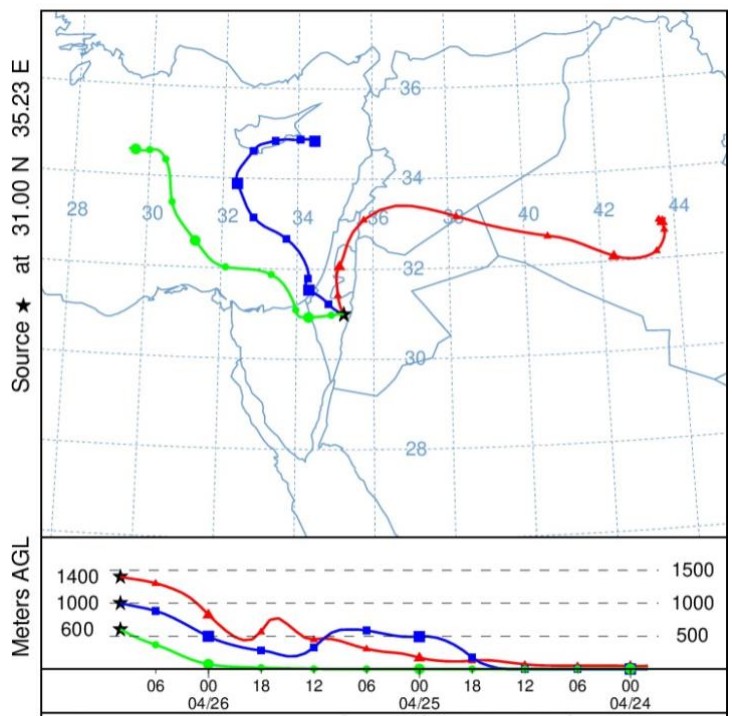

**Figure 6: Sixty hours back-trajectory of the air mass reaching the Tzafit Basin at April 26, 2018 10 UTC at heights 600, 1000 and 1400 m AGL.**

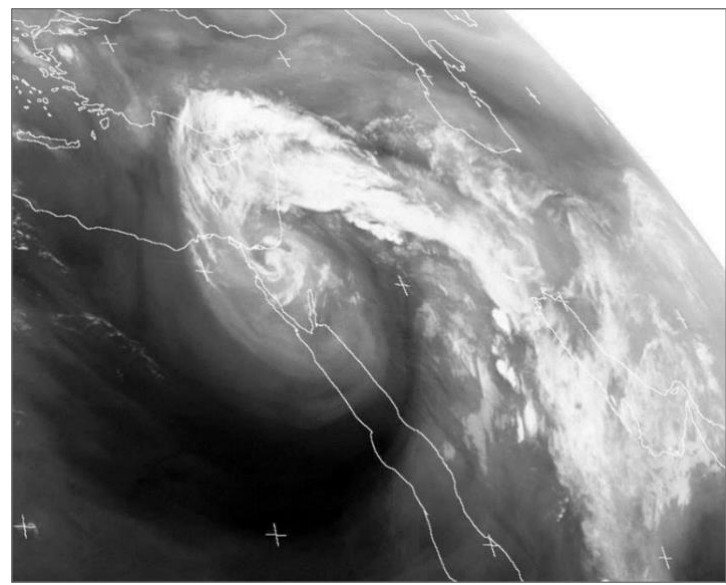

**Figure 7: METEOSAT Water Vapor channel 6 (6.85–7.85 μm) imagery from April 25, 2018 21 UTC showing the cloud strip entering Israel from the east and the vortex over the south-eastern Mediterranean.**

### 3.2 April 26: main rain episodes

In the early morning of April 26, the northern part of Israel was the only one receiving rain, mostly moderate, originating from stratified cloudiness (based on hourly radar images, not shown). After several hours of fair weather, at noon hours, three major rain centres developed. Figure 8 shows the three centres as reflected by two

observational means. i.e., satellite IR and radar imagery, and two diagnostic tools, vorticity field at a 30 km resolution and MKI at a 2.8 km resolution.

The first rain centre, composed of several cells, was most active between 0900-1030 UTC and produced the Tzafit flood. This centre yielded up to 50 mm rainfall, with a maximum intensity of 180 mm/h, based on 5 minute averages of radar measurements. This intensity has a return period of > 75 years for the pertinent region (Rinat et al. 2020).

This centre can be noted in the radar image for 09 UTC (Fig. 8c). Its evolution is represented by a series of radar images (Fig. 9), showing that it resulted from the passage of successive 3 rain cells that crossed the region from north to south, with an average speed of 12 ms$^{-1}$. Figure 8e indicates the existence of a positive vorticity anomaly in the 500 hPa level over the Dead Sea (denoted in blue, north-east of Tzafit), with ~50 km diameter, in 09 UTC, implying that positive vorticity advection existed over Tzafit via the north-easterly flow at that level.

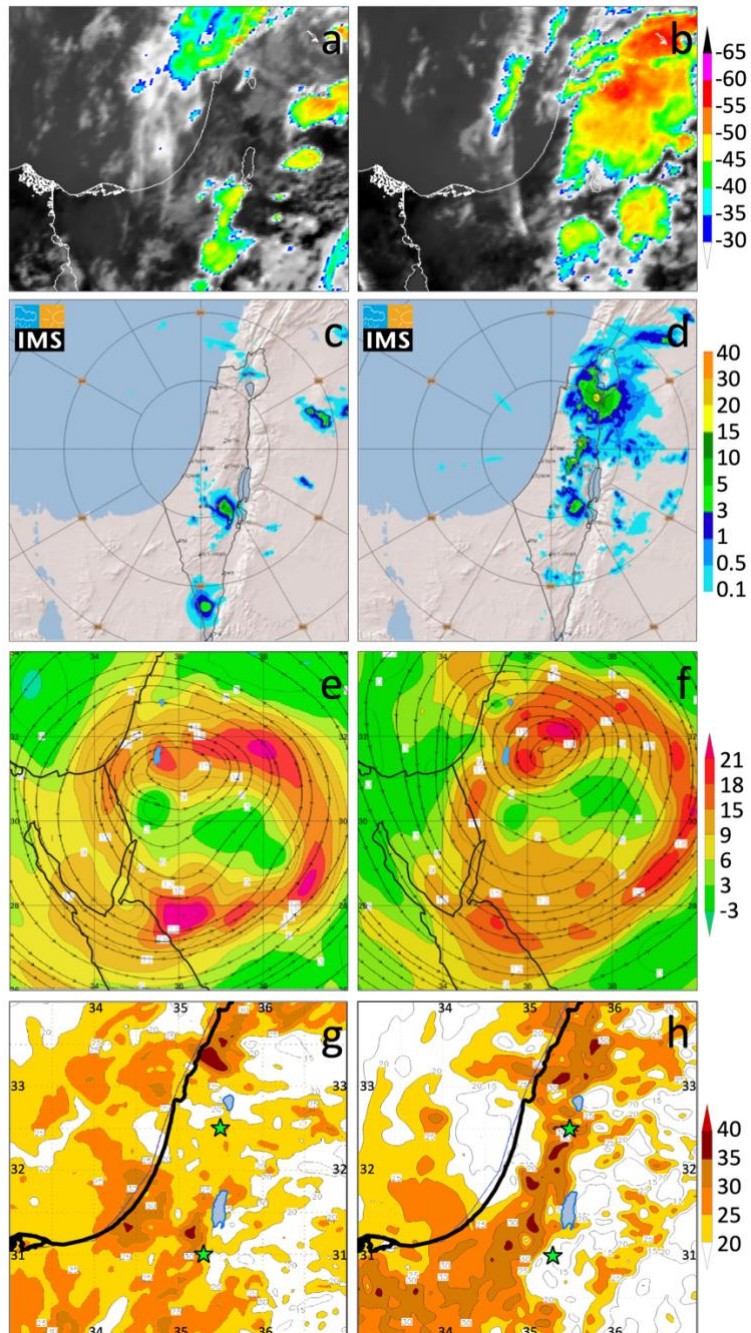

**Figure 8: April 26 imageries for 09 (a,c,e,g) and 12 UTC (b,d,f,h) corresponding to the major rain centres including MSG IR images (a,b), one hour forward integration Radar [mm h⁻¹] images (c,d), relative vorticity [s⁻¹×10⁵] and streamlines at 500 hPa (e,f), and MKI [°C] (g,h). Panels e-f are based on ERA5 data at 30 km spatial resolution and panels g-h are based on COSMO model at 2.8 km spatial resolution.**

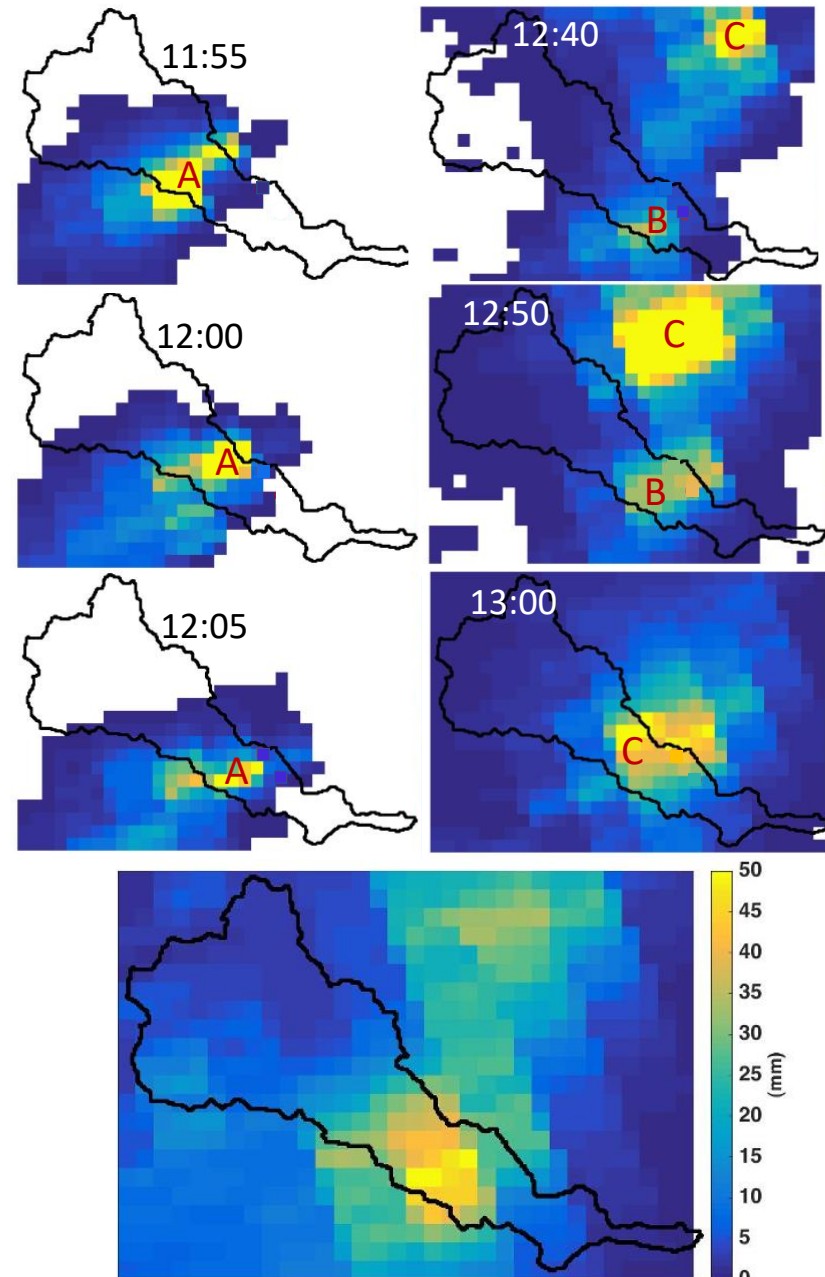

**Figure 9: a. A series of radar images indicating the progression of rain cells (denoted A, B and C) that crossed the Zafit basin (1 pixel is equivalent to 0.25 km². The time notations refer to the summer clock, UTC+3h. The units are mm/h, based on the conversion method of Mara and Morin (2015). b. Total rain amount accumulated between 11:40 and 13:40 LST. Note the meridional orientation of the rainfall maximum, pointing at the "training effect" (with the curtesy of Yair Rinat).**

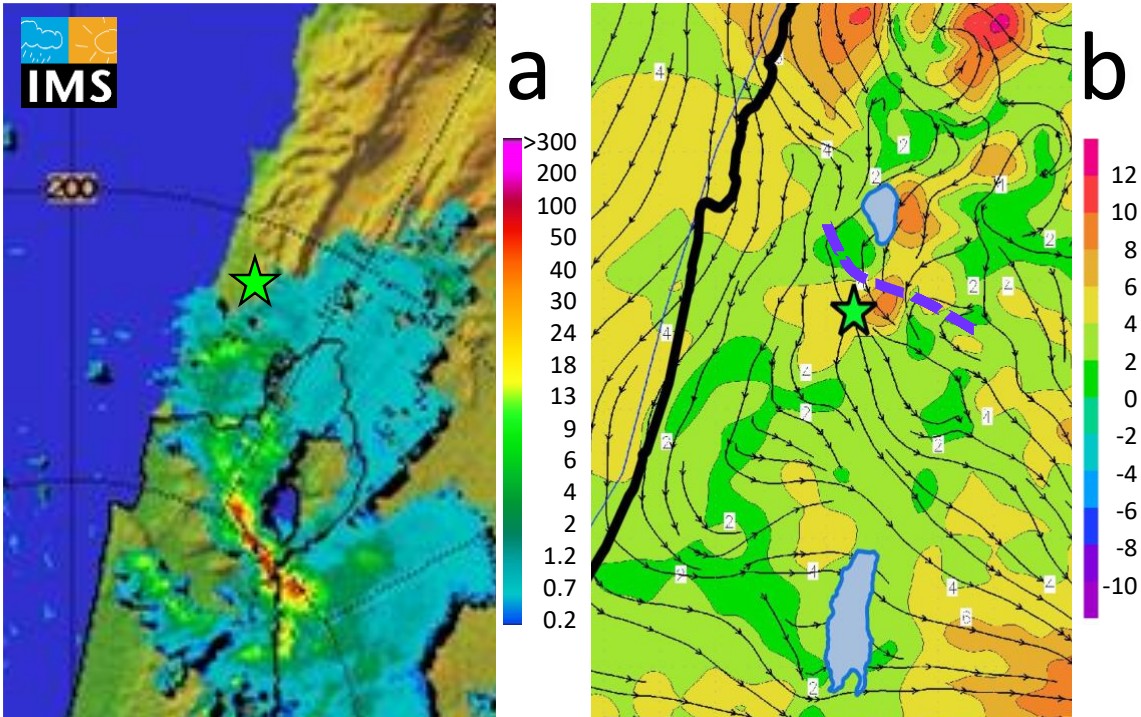

**Figure 10: a. Radar image (in mm/h units), indicating the precipitative element that crossed Beit Shean (Lake Kinneret, seen northeast of it). Note that the maximum rain intensities reach 50 mm/h. b. 850-hPa streamlines and wind speed, with a purple curved line denoting the trough line. Beit Shean is denoted by a**
275 **green star in both parts. The radar image is from 12:20 UTC and the wind map from 12:00 UTC.**

Later on, around 12 UTC, another rain centre developed in the Jordan valley (near the city of Beit Shean, 32.5°N 35.5°E) at north Israel, in a form of a cloud line, as reflected by the satellite and radar images (Fig. 10a). This cell yielded rainfall of 30-45 mm, with a maximum of 72 mm (record breaking since start of record at 1943). The
280 maximal 10 min averaged rate was 116 mm/h, having a return period of 100 years (Porat et al., 2018). During this rain episode, a lower-level trough extended toward the region from the southeast (i.e., Jordan, Fig. 10b). Its curved shape and orientation (denoted by a purple line) resemble these of the cloud line. This system crossed the region southward at an average speed of 7 ms$^{-1}$. The rain in Beit Shean can also be attributed to positive vorticity advection at the 500 hPa level, caused by north-easterly winds ahead of a pronounced core of positive vorticity anomaly of
285 $> 2\times10^{-4}$ s$^{-1}$ and ~100 km diameter, centred over western Jordan (32.3°N, 36.7°E). This core approached the region (red point in Fig. 8f) from the east and was one of the two vorticity centres that constituted the synoptic scale cutoff low at 12 UTC (Fig. 2).

The third rain centre started to develop at 12 UTC and lasted ~ 2 hours. This centre was aligned with the water divide of the Samarian and Judean Mountains and its adjacent slopes facing the north-westerly winds (Figs. 8b, d), and therefor can be considered as forced by orography. This rain centre yielded 30-35 mm, with 10 min averaged maximum rate of 100 mm/h in Jerusalem, having a return period of 100 years (Porat et al., 2018). In the late afternoon, southern Israel was still under precipitative cloudiness (not shown).

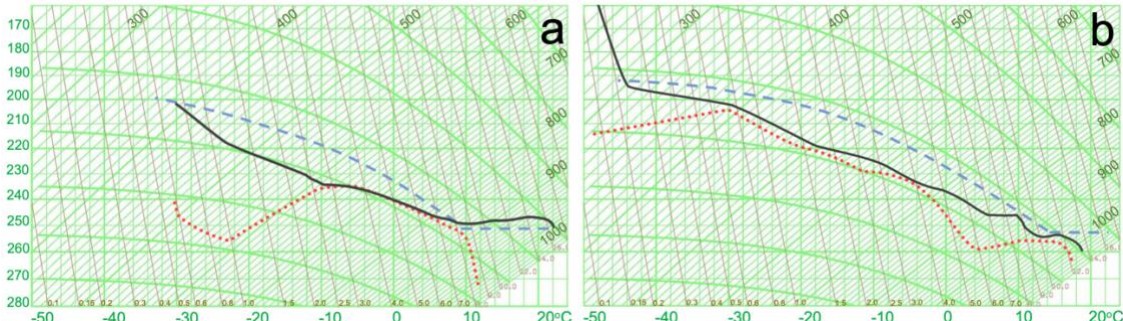

**Fig. 11: Vertical profiles for (a) Tzafit (26 April 2018 0930 UTC) and (b) Beit Shean (26 April 2018 1230 UTC). The black lines denote temperature, the red dotted lines – dew point and the blue dashed lines show the adiabatic cooling of air parcels ascending from the surface.**

Vertical profiles for two major rain centers, in Tzafit (Fig. 11a) and near Beit Shean (Fig. 11b), derived from the fine resolution data of the COSMO model (described in Sec. 2). Each one includes the temperature, dew point and a line denoting the cooling of an air parcel that is lifted diabatically from the surface. Both reflect moist and conditionally unstable conditions. The lifted air parcel in both of them is warmer by ~5°K than its surrounding temperatures within layers of several Km width. This implies that in both rain events the ascending air currents within the rain cells were subjected to intense buoyancy forces and, hence, had high vertical speeds. Accordingly, the CAPE and the MKI over their locations at the time of their occurrences exceeded 1200 J kg$^{-1}$ and 30K (1740 in Beit Shean), respectively, values that are favourable for thunderstorms. Moreover, the tops of clouds of the three precipitative centers reached -50°C. According to the temperature profiles over the study region during the storm, this implies that these cloud tops exceeded an elevation of 9000 m.

## 4 Discussion

Precipitation extremes in the Mediterranean Basin are usually induced by large-scale atmospheric circulation, namely low-pressure systems with large pressure gradients (Jacobeit et al. 2017). Therefore, the analysis starts from the synoptic system, with which this storm was associated, namely the upper level cutoff low. This cutoff low

entered the EM from northwest, deepened and crossed the Levant at 30°N latitude (Figs. 2, 3). The maximum vorticity measured in 500 hPa level was $> 7.2 \times 10^{-5}$ s$^{-1}$, which is comparable with the value of $5 \times 10^{-5}$ s$^{-1}$ at the same level found by Flocas et al. (2001) for the winter season in the Mediterranean. Apparently, this implies that the synoptic factor was dominant in rain production. However, convective nature of the rain and spotty distribution was demonstrated by radar images (Figs 8c, d) and was even reflected by the rain map of the entire storm (Fig. 1). Following the quasi-geostrophic approach, most convective outbreaks occur in broad south westerly flow aloft ahead of an approaching trough (Doswell, 1987). In April 26, under the north westerly flow behind the upper-level cyclonic system, its direct dynamic supporting effect on rain formation is expected to be weak, or even negative (Fig. 2c). At this sector, negative upper-level vorticity advection and lower-level cold advection are expected, both implying subsidence in the mid-levels. Inspection of the lower-level temperature and wind fields (not shown) indicate that at that stage, the cold core associated with the upper-level cyclone was centred over the Negev Desert. The lower-level winds over the study region were parallel to the isotherms, so that no temperature advection existed. Similar configuration existed with respect to the upper-level vorticity field, implying that vorticity advection on the synoptic scale also did not take place there. Moreover, the omega field at the mid-levels in the synoptic scale over the region was near zero. The above implies that the synoptic scale dynamics did not have a direct effect on the rain formation on that day.

The major synoptic factor that directly contributed to the rain formation in this storm is the wind. One implication is the onshore moisture transport, accompanied by an uplift imparted by its encounter with the coastline and later on, with the mountain ridges. The other is the upper-level cold advection, leading to thermal instability, which is further elaborated below. Actually, only a small fraction of the rainfall in this storm was orographic (over the Judean Mountains), whereas the majority of the rain was observed far from the coastline and beyond the water divides of the mountain ridges (Fig. 1, note the divide line).

The meso-scale analysis performed exposes several uplifting mechanisms, not captured by the synoptic-scale data, which can explain the occurrence of the Tzafit and Beit Shean major rain centres. Each of these two centres was found to be associated with a positive vorticity advection ahead of a positive anomaly in the 500 hPa level in the ERA5 maps with 30×30 km resolution. The first anomaly was located east of Tzafit at 09 UTC, prior to the rain there (Fig 8e), and the other, east of Beit Shean at 12 UTC, when the rain system started its movement southward over that region (Fig. 8f). The third mechanism was identified in the wind field of the COSMO data at the 925 hPa level, in the form of a lower-level trough (Sec. 3.2, Fig. 10b), with shape, orientation and progression velocity similar to these of the rain producing system (Fig. 10a).

Despite the affinity of the Mediterranean cyclones to mid-latitude cyclones, due to the limited moisture sources in the Mediterranean Basin (Ziv et al. 2010), the rain they produce highly depends on moisture supply, rather than on the presence of conveyor belts or front. The lower-level system in the storm analysed here, can be considered as a

345 'Syrian low' (Kahana et al., 2002, see Sec. 1 above), which belongs to the Mediterranean cyclones. The Syrian low resembles the 'deep low to the east', which is one of the 7 types of Cyprus low defined by Alpert et al. (2004). Saaroni et al. (2010) showed that most of the rain associated with the 'deep low to the east' is distributed over the coastal regions and the western slopes of the Judean Mountains, facing the offshore north westerly winds blowing from the Mediterranean. This indicates that the Judean Mountains, extending up to 800-1000 m, block effectively

the moisture, which is presumably, concentrated in the lower-levels. The heavy rains that were observed inland in this storm, and the signature of non-orographic major rain cells in the rain maps indicate that the moisture sources were not limited to the lower-levels. The presence of moisture at the mid-levels is deduced from back-trajectory arriving at Tzafit at the time where the flood occurred (Fig. 6). This show a band of mid-level moist air that originated east of the Levant, turned cyclonically around the cyclone centre and entered the region from the west.

In the EM, rains associated with Mediterranean cyclones, as in the case studied here, are convective in nature. In the winter season, cold air originating from south Europe moves over the warmer Mediterranean water and becomes unstable before entering Israel (Shay-El and Alpert 1991). In the late spring, Europe becomes warmer, whereas the Mediterranean remains cool, due to its lagged response to the annual cycle, so that the passage of a Cyprus low over the EM does not necessarily lead to instability. In the case studied here, the instability can be attributed to a

negative temperature anomaly in the upper levels (in the order of -4 K in 500 hPa, not shown) that covered the southern Levant, as a part of the cutoff low. A tendency of the rain in the Negev Desert to be convective, especially in the transition seasons, has been noted previously. Sharon (1978) pointed at a gradual increase in the proportion of localized showers in the transition from the Mediterranean part of Israel further into the more arid south. Systematic evidence for this tendency is well reflected in the low spatial correlation obtained among several rain

stations for April over southern Israel (Kutiel, 1982).

The considerable rainfall in the three major rain centers described above can be explained by several factors:

a) The high degree of instability, originated by the upper-level cold air, which was further enhanced by the intense solar radiation prior to the three rain events, as implied by the date, latitude and the clear sky (Fig. 8a). This effect is reflected by the MKI maps (Figs. 8g,h), showing much higher values inland compared to these along

the coastal region (35°C in Tzafit Basin compared to 13.2°C in Beit Dagan) and by a gradual increase in CAPE

during the morning hours (not shown), up to 1740 Jkg$^{-1}$ in Beit Shean. This factor can explain the extreme rain intensities observed in the three major rain centers.

b)  The high elevation of the cloud tops in the three major rain centers, exceeding 9,000 m ASL., 2,500 m higher than that typifying the tops of the winter thunderclouds in Israel (Altaratz et al., 2001).

c)  For the convective rains observed over the Judean Mountains at the noon hours, a central factor is orography.

d)  The above 3 factors led to extreme and exceptional instantaneous rain rates, exceeding a 75 years return period in Tzafit and a 100 years in the other two major rain centers.

e)  A significant contribution to the high rainfall in the Tzafit basin can be attributed to the propagation vector (train effect), i.e., repeated areas of rain cells that move over the same region in a relatively short period of time, which may cause flash flooding (e.g., Cordifi et al., 1996).

f)  An optional factor that may explain the repetitive formation of rain cells north of Tzafit is a mountain-valley circulation (anabatic) uplift over the eastern slopes of the ridge that extends from Judean mountains southward (Fig. 1). This could be expected in light of the clear skies prior to this event, but did not find any signature in the output of the COSMO model.

g)  The high rainfall in Beit Shean can be partly explained by the slow movement of the rain system, at a speed of 7 ms$^{-1}$.

Beyond the major direct contribution of the instability, moisture supply and topographic features to the evolution of the major rain cells, the role of the synoptic factor, namely the cutoff low, can be considered as a supportive background for these effects. The impact of cutoff lows on rainstorms over the Mediterranean has also been shown by Porcu et al. (2007) and by Fragoso and Tildes Gomes, 2008 (Sec. 1). This goes in line with the fact that 11 out of the 12 reference rainstorms that occurred over the study region in the late spring were accompanied by upper-level closed cyclones.

The slow eastward propagation of the cutoff low exacerbated the severity of the storm by extending its duration. The typical translation speed of Mediterranean cyclones is 5-10 m s$^{-1}$ (Alpert and Ziv, 1989). The speed of the cutoff low was 7-9 m s$^{-1}$ while moving along the Libyan coast, but slowed down to 3-5 m s$^{-1}$ while approaching the EM (Fig. 3). The series of 300 hPa PV maps (Figs. 5a-c), on which a conceptual sketch of a dipole type is superposed, consisting of a blocking high located north of the cut-off low, follows Yamazaki and Itoh (2013). These charts show a lobe of negative PV (blue) over the Balkans, north of the cutoff low, lingering over the Nile delta. The power of PV maps stems from the feature of PV anomalies (in the spatial sense) to induce circular flow out of their boundaries, i.e., cyclonic for positive and anticyclonic for negative. The negative anomaly centred over Eastern Europe exerted an anticyclonic flow over the region, which produced easterly winds south of it, including

the Levant, where the positive anomaly (with which the cutoff low was associated) was located. At the same time, the positive anomaly over the southern Levant exerted a cyclonic flow around it, including easterly winds over Eastern Europe, where the negative anomaly was positioned. This is most clearly demonstrated in Fig. 5c. The outcome of the superposed effects is that both anomalies were subjected to forces acting against the mid-latitude westerlies. This explains the slow progression of the cutoff low, while reaching the Levant.

## 5. Summary

An intense rainstorm hit the Middle East between 24 and 27 April 2018, produced heavy flash floods in Israel that took the lives of 13 people. Ten of them lost their lives in April 26, in a flash flood that resulted from rain showers with intensities of > 75 years return period. The study describes the major rain systems affecting Israel in that day, and analyses the atmospheric processes leading to their formation. The dominating synoptic feature was a cyclone in all levels that crossed Israel in its way eastward. The rain distribution for the entire storm was not evenly distributed. Three major rain centres were active during April 26. One, which caused the fatal flood, developed over a relatively flat terrain, the second developed upslope the mountains, i.e., orographic, and the third formed in the Jordan valley. The synoptic factor supplied the background for the rain-storm and the meso-scale features determined the locations and times of the major rain events.

The synoptic background of the storm was an upper-level cutoff low that originated from south Europe and slowed its movement while approaching the Levant. This cutoff low was found to have the highest vorticity with respect to the set of 12 'reference storms' observed over the study region in the late spring during the latest 33 years. The lower levels were dominated by a cyclone, centred east of Israel. The implied winds were north-westerly, hence advecting moist air from the Mediterranean inland. During the approach of the upper-level cyclone, the atmosphere over the study region became conditionally unstable, with instability indices reaching values indicating potential for thunderstorms. At the same time, the precipitable water increased by a factor of 2 over the study region, which triggered deep moist convection from April 25 onward. The mesoscale distribution of MKI (Modified K index) indicated that the instability was considerably higher inland than along the coastal plain. This may explain why most of the rain was observed there. In addition to the lower-level moisture advection, a mid-level band of moist air, extending from Iraq westward, curved cyclonically through Syria and the EM and entered south Israel from northwest. The combination of both moisture contributions and instability can explain the severe rain showers over Israel observed during April 26.

The present study raises several questions for further research. One of them concerns the involvement of closed upper-level cyclones as a major factor for severe rainstorms over the Negev Desert in the late spring. The spring season in the Mediterranean is characterized by a general weakening and reduced frequency of Cyprus lows. Moreover, the late spring is the time in which the descending branch of the Hadley cell takes over the Mediterranean. In light of the above, the specific conditions that lead to intensification of Rossby waves that lead

to their breaking into closed cyclones should be investigated.

The most intense showers associated with the storm studied here were observed while the study region was situated west of the upper-level cyclone, where negative vorticity advection is expected. Similar situations were noted by Kahana et al. (2002) in the framework of the 'Syrian low', which produced major floods in the Negev Desert. A similar situation was analysed by Morin et al. (2007), which resulted in a fatal flood in Israel, between 31 March

and 1 April 2006.

Rain production in convective storms requires the coherent action of many processes acting over broad ranges of space and time scales. In order to provide an adequate description of the factors involved and to get warnings for potential flash-floods, such as the one that occurred in Tzafit, there is a need to identify key mechanisms that can only be identified by state-of-art regional models, such as the COSMO model, used here.

*Acknowledgements*:

The authors wish to thank Efrat Morin and Yair Rinat from HUJI for the hydrometric data and the radar images. We are grateful to Noam Halfon and Yoav Levi from the Israel Meteorological Service (IMS) for providing the IMS C-band Doppler radar imagery and the integrated rain maps and to Elyakom Vadislavsky for the MSG IR

images and for his assistance and scientific inputs while running the IMS – COSMO model and to Michal Kidron and Guy Keren from HUJI for their help in the preparation of the figures. BZ thanks the Israeli Science Foundation (ISF, grant number 1123/17). We also wish to express our gratitude to the three anonymous referees for their most helpful comments and constructive suggestions which led to substantial improvements of the paper.

*Data availability*. ERA5 ECMWF reanalysis data have been downloaded from the Copernicus Climate Data Store (https://cds.climate.copernicus.eu, Copernicus, 2020). Meteosat Second Generation water vapour imageries were retrieved from EUMETSAT data centre (https://navigator.eumetsat.int/product/EO:EUM:DAT:MSG:HRSEVIRI). Air back-trajectories for detecting moisture transport were retrieved from the site of NOAA HYSPLIT MODEL (Stein et al., 2015, https://www.ready.noaa.gov/HYSPLIT.php)

*Author contributions*. UD conceived the study, set the goals of the article, and developed its methodology. IML, selected and analysed the data sources. BZ was engaged in the dynamic aspects of the study and their issuing calculations. PK performed the numerical simulation with IMS – COSMO model. All the authors contributed to the interpretation of the results and to the manuscript writing.

*Competing interests*. The authors declare that they have no conflict of interest.

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
