# Peer review of "Atmospheric Conditions Leading to an Exceptional Fatal Flash Flood in the Negev Desert, Israel"

_Natural Hazards and Earth System Sciences, 2020_

## Referee Comment (RC1) · Anonymous Referee #1 · 24 May 2020

1. General comments The study "Atmospheric Conditions Leading to an Exceptional Fatal Flash Flood in the Negev Desert, Israel" by Uri Dayan, Itamar M. Lensky and Baruch Ziv presents a thorough analysis of the synoptic and sub-synoptic patterns which produced a major storm in southern Israel, causing the death of 10 people. To my view, the significance of the paper is clear, given that there seems to be no sufficient knowledge to alert people ahead of such storms. Although analyzing one case study, the study has a substantial contribution to the understanding of flood-bearing rainstorms in Israel and its vicinity, as general conclusions are drawn and using a comparison to another eleven spring storms that caused floods in the area. I find the analysis done along the cyclone track (MCV, speed) especially interesting, as it shows this cyclone intensified over Israel, and was therefore specifically intense over this region,

and as it gives sort of reference values that can help in forecasting of severe storms. Furthermore, I think that the study of the atmospheric conditions during flood-bearing storms is much needed as these are the biggest natural hazard causing fatalities in the region. The scientific and presentation quality is good, although I have quite a few comments on both topics. Overall, I think that this study is interesting and important, and is worth publication in NHESS after addressing the points raised below. Three of my comments worth specific attention and are written under "specific comments". Other comments are smaller in nature (even if some are also conceptual) and most of them are technical in nature.

2. Specific comments - L160-174: These lines present an interesting idea; however, I am not sure the support given to it is enough. It is not clear to me if the slowing of the cyclone occurs every time there is a closed cyclone over the region or only when there is a dipole pattern. If the dipole is needed, I am not sure whether the 500 hPa GPH anomaly can explain the dipole. The way I understand it from your description is that both the southern part of the high and the northern part of low induce easterly flow and slow the propagation of both systems. However, examining the actual synoptic maps (and not the anomaly maps), I cannot see the highs. I can observe two ridges, one from to the west of the low and the other to the east of it (extending in Fig. 2c to the north of the cyclone). Do you account the ridge from the east for the slowing of the systems? I am not sure the pressure gradient is represented well enough by the anomaly maps. - L189-196: What is the timing of the rainfall that produced the flood? Rainfall is observed in the region of "Tzafit" both on 06-12UTC and in 12-18UTC, while significant omega is only seen in 15UTC. Please address this point, as it is crucial to understand whether ERA5 omega values are indeed accurate enough to capture such a small-scale rain spot. - L225-236: I am wondering about the Tropical moisture source. Fig. 8 as well as Fig. 2c (and to some degree 2b) show southwesterly flow to Saudi Arabia, all consistent with your "tropical" source attribution. However, examining Fig 2a (and 2b) I can see convection triggered close to the cyclone's center, with northerly (Mediterranean) flow at low elevations. This makes me think the moisture source could

have been Mediterranean to begin with, and this moisture (combined with the deep convection along the cyclone's track) could be the only moisture source to precipitation from this cyclone. The southwesterly flows in Saudi Arabia (Fig. 8 and Fig. 2c) existed probably at the night of the 25th of Apr, but to reach this region with sufficient moisture, you would probably had to look for southwesterly flows prior to this night, and this does not seems to be the case (Fig. 2a and b). Moreover, 18-24 h before arrival to "Tzafit Creek" the position of the red backtrajecktory is in southern Syria. Going further back in time with the red trajectory gives Iraq (ground level) as the origination of the track. I am not familiar enough with this region, but I would expect it to be rather dry, and anyway, the track does not go all the way back to tropical latitudes. Please add some support for your moisture source determination, or correct this part of the text, as I suspect the Mediterranean could possibly be the only source in this case. One possible solution is to plot moisture content along the backtrajectory, and another one could be plotting a matrix of trajectories arriving to close-by locations.

3. Technical (and more specific) comments - L10: Please state either that the ten casualties were among the people that die during the event (to the best of my knowledge, three more people died in other streams in Israel during the same event) or that among the flash floods one was specifically deadly, killing 10 people. - L11-12: "The timing of the storm is also unique, at the end of the rainy season, when rain is relatively rare and spotty". Is it unusual to get extreme storms at the transitional seasons at the Negev desert? If not, please correct. - L13: Please rephrase "one of the latest spring severe events in the region during the last 3 decades" to "one of the most severe latest-spring events in the..." or something in that spirit. The same holds for L46-47. - L17: consider rephrasing "latest" to "last". - L36-37: consider rephrasing "most weather-related fatal hazard" to "most fatal weather-related hazard". - L40-44: It is not clear whether most rainfall in the desert areas occurs during Dec-Feb or only over the non-desert parts of Israel. Please clarify this point. - L55: "over which floods took place" during this event? In general? Please explain. - L58: Please explain how "maximum intensity" is defined. - L58: Can you please elaborate on the 11 storms? Are these the only storms during

Apr-May over this period? Are these the highest discharge storms (as suggested in L66)? Are these the same storms listed in Table 1? If these are the same storms, please number the different storms in the list, as it is hard to find 11 storms in it. If they are not the same storms, please list these storms in a table as well. - L61: what is the spatiotemporal resolution of the radar data used? If I am not mistaken, Marra & Morin (2015) data come from another radar in the region. - Figure 1: It is not clear whether this map comes from accumulated IMS radar data ("RADAR" is stated in the figure's caption) or from the "rain measuring network" (L60). The shape of the contours looks as if coming from an interpolation of gauges. Please explain and change the text referring the figure or the figure's caption accordingly. It seems radar data is more appropriate to use since rain gauges in the analyzed region seems scarce. - L66: What do you mean in "the data of floods"? Does this refer to hydrographs? Peak discharge? - L70: please add a reference to this statement. Consider one (or more) of the following for this specific region: (Armon et al., 2018; Dayan & Morin, 2006; Kahana et al., 2002, 2004); and in general: (Bárdossy & Filiz, 2005; Borga & Morin, 2014; Doswell et al., 1996). - L69-74: please explain why you use different datasets (ERA5 and NCEP/NCAR). Do coarse scale ERA5 data the same (or very similar) to NCEP/NCAR data? If not, the use of ERA5 data both in the synoptic scale and in the meso-scale is probably better. - L83: What are the units of the cyclone depth? is it m or hPa? Do you measure the depth at the 500 hPa as well? Does V refer to the 500 hPa? - L90: "usually 500 hPa" – is this the case here or do you consider another pressure level? - L98-103: did you consider using the Beit Dagan sounding data for the MKI index as well? This could help understanding whether there are significant differences in ERA5 data in respect to the sounding data, which is not situated within your study area. - L104-105: Where does PW come from? Is this based on ERA5? Please indicate it in the text. - L107-109: Can you describe what kind of backtrajectories did you used? Which atmospheric model was used (NCEP/NCAR reanalysis / ERA5 / Other data)? What kind of levels did you use (model / pressure / isentropic)? - Fig. 2: Could you please describe the MODIS imagery? Does the image come from

12UTC as well? Is it from a specific MODIS (Terra/Aqua)? If so, please write it. - L131-149 and Fig. 3: It is not clear to me how did you track the cyclone at the 500 hPa. Was it through the large scale NCEP/NCAR data or the ERA5? Or possibly using satellite images? Please explain it either in the materials or specifically when referring to the cyclone's movement. - Fig. 4: Please correct the caption ("with maximum of K..."), consider choosing a more appropriate colorscale (centered at zero or starting at zero), a better in-figure-caption units (currently it is "10ˆ1 K m**2 kg**-1 s**-1"), and possibly use PVU units instead. - L141: "This is rather exceptional..." – did you compare the value to other values from your reference period? This could be interesting and could also put the results in a broader context. - L185 please replace the square sign with a degree sign. - Figure 7, middle panels: Do these panels represent radar, gauges or a combination of radar and gauges? Panel f seems to be some combination of gauges and radar, but this is not clear from the caption or "materials" section. L181: what do you mean by "integrated"? Does this refers to the combination of sources or to the accumulation of rainfall? Please add units to the MKI legend or caption. - L194: "-10 Pa s-1" do you mean -1 Pa s-1? If not, please correct Fig. 7's legend. - L244: Please see previous comment about the "unique timing" of the storm. - L245: This paragraph seems to me as a summary paragraph, however the term "Syrian low" was only introduced in L245. Could you please either remove this line from the summary or describe this term in the introduction/results sections? - L285: Cape values in Fig. 2 reach to 909 and not to 1000. Please correct or explain this notion. - L286-289: Contribution of tropical moisture was shown in a number of studies; however, I am not sure this is the case here. Please address the previous comments about this subject. If you are certain about the tropical origins, please add more references or add "e.g.". - L290-294: The point raised here is interesting and fits well a discussion section. Since this section of the paper is written in bullets, it looks like a summary or a conclusion. Please give a better distinction between the summary and the discussion. This will help readers to understand which parts in this section should only be treated as a discussion and are therefore introduced firstly in this section, and which are summarizing prior sections.

Perhaps the discussion can remain as one section and the summary can go with the conclusions, and in this way you will reduce redundancy. - L299: Please either add "not shown" after the "-5 K" or show this anomaly. Currently, there is an impression this anomaly is given in Fig 6. - L303-L310 "unique intensity", "exceptionally severe rainstorm", "severity of this rainstorm": what exactly all of these expressions refer to? Do you mean the MCV? The rainfall (is it really that rare)? The casualties (were they influenced by the extreme severity, bad luck, or bad circumstances)? Please explain this point, and better to do it in previous sections of the paper. - L313-314: Is this the case here? Was it written previously? - L315-317: Please see previous comments about tropical moisture origins (although this conclusion seems reasonable anyway). - L318-320: Please also see previous comment regarding L189-196.

4. References Armon, M., Dente, E., Smith, J. A., Enzel, Y., & Morin, E. (2018). Synoptic-scale control over modern rainfall and flood patterns in the Levant drylands with implications for past climates. Journal of Hydrometeorology, 19(6), 1077–1096. https://doi.org/10.1175/JHM-D-18-0013.1 Bárdossy, A., & Filiz, F. (2005). Identification of flood producing atmospheric circulation patterns. Journal of Hydrology, 313(1–2), 48–57. https://doi.org/10.1016/j.jhydrol.2005.02.006 Borga, M., & Morin, E. (2014). Characteristics of Flash Flood Regimes in the Mediterranean Region. In N. Diodato & G. Bellocchi (Eds.), Storminess and Environmental Change Climate Forcing and Responses in the Mediterranean Region (pp. 65–76). Dordrecht: Springer Netherlands. https://doi.org/10.1007/978-94-007-7948-8_5 Dayan, U., & Morin, E. (2006). Flash flood – producing rainstorms over the Dead Sea: A review. New Frontiers in Dead Sea Paleoenvironmental Research: Geological Society of America Special Paper, 401(04), 53–62. https://doi.org/10.1130/2006.2401(04). Doswell, C. A., Brooks, H. E., & Maddox, R. A. (1996). Flash Flood Forecasting: An Ingredients-Based Methodology. Weather and Forecasting, 11(December 96), 560–581. https://doi.org/10.1175/1520-0434(1996)011<0560:FFFAIB>2.0.CO;2 Kahana, R., Ziv, B., Enzel, Y., & Dayan, U. (2002). Synoptic climatology of major floods in the Negev Desert, Israel. International Journal of Climatology, 22(7), 867–882. https://doi.org/10.1002/joc.766 Kahana,

R., Ziv, B., Dayan, U., & Enzel, Y. (2004). Atmospheric predictors for major floods in the Negev Desert, Israel. International Journal of Climatology, 24(9), 1137–1147. https://doi.org/10.1002/joc.1056 Marra, F., & Morin, E. (2015). Use of radar QPE for the derivation of Intensity–Duration–Frequency curves in a range of climatic regimes. Journal of Hydrology, 531, 427–440. https://doi.org/10.1016/j.jhydrol.2015.08.064

---

## Referee Comment (RC2) · Anonymous Referee #2 · 5 Jun 2020

General comments

This manuscript deals with a very interesting case of a high-impact storm. There were floods in the Negev desert in southern Israel, in which unfortunately 10 people died. The event had not been expected as it hit at the end of the cold season.

A scientific investigation of this storm is important in order to understand the underlying processes and to improve the forecast for such flooding events. Additionally, a climatological classification as carried out in the work, helps to better estimate the potential of these events.

However, the work is essentially limited to large-scale processes that are also not completely discussed. In addition, there are attempts to connect small-scale processes to

the synoptic scale, which fail because of the separation between dynamics and thermodynamics and not least because of the selection of data and the way these are presented. Finally, the manuscript gives repeatedly undisputed hypotheses, as well as some inconsistencies.

The main goal of this work as given in lines 50-52 ("In section 3 we describe the event and identify the unique dynamic and thermodynamic conditions that lead to the severe convection, as well as the sources of moisture for the rain formation in this storm.") needs to be elaborated more, especially with respect to across-scale processes with respect to rain production. Evidence needs to be given to the hypotheses that are presented.

Specific comments

The manuscript is focused on the analysis of the large-scale flow, which is compared with similar flooding events. The intensity and track of the corresponding cut-off low in 500 hPa is unusual for the season. Parts of the work, however, disagree on whether the event is a typical weather situation, with flooding occurring outside the rainy season in the desert or whether it is a unique situation (compare lines 42-45: "The rest of the annual rainfall occurs in the transitional seasons and is contributed by precipitating tropical synoptic-scale systems and by Cyprus Lows. A significant part of them occur in the desert areas and are characterized as intense rain events of small spatial extent and short duration, some of which produce flash floods (Kahana et al. 2002; Dayan and Morin 2006, Greenbaum et al. 2010)." and lines 245-247: "The location of the surface cyclone was similar to the 'Syrian low', defined by Kahana et al. (2002) as one of the major systems causing floods in the Negev Desert." with line 51: "unique dynamic and thermodynamic conditions"). The impression is that the large-scale weather conditions are well known for flooding, whereas only the temporal appearance and the intensity and coverage were unique. Readers unfamiliar with the given weather patterns, i.e. 'Syrian low' and 'Cyprus Lows', can be confused which of these two is mainly associated with flooding events. It would be good to explain these weather patterns at the

beginning of the text in some detail, e.g. mid-level flow and surface pressure field.

Apart from this analysis of mid-level and SFC flow, hypotheses are (repeatedly) raised that are not discussed further:

Lines 311-312 "The combination of a cut-off low with small radius and large hypsometric depth, implies high curvature relative vorticity, with strong dynamical forcing on rain formation." At this point, the authors need to be more precise. "Strong dynamical forcing" refers to quasi-geostrophic processes, and large-scale lift is the order of cm/h. Does QG lift affect rain formation directly? Moreover, only the contribution by differential cyclonic curvature vorticity advection to QG lift is mentioned. Since the weather charts indicate north-westerly flow, the reader may wonder if a cold air advection maximum may cancel QG lift in a similar event. And what about shear vorticity?

Lines 193-195: "It should be noted that the three rain centers are located within a region of negative Omega (ascendance, Fig. 7c), with an extremum value of -10 Pa s-1. This implies that these rain systems are dynamically supported." In this manuscript, the "dynamic factor" (quasi-geostrophic lift) is analyzed separately from the "thermodynamic factor" (thunderstorm) (lines 123-125: "Two complementing factors contributed to the rain formation. One is dynamic, i.e., vertical ascent, associated with the cyclonic system described below (Sec. 3.2). The other is thermodynamic, which is composed of instability and moisture supply, described in Sec. 3.3."). There are two criticisms to approach. First, the omega fields show a combination of lift on different scales, including convection parametrization. It is therefore not possible to conclude that the given omega fields indicate just dynamic lift. Secondly, both factors influence each other, so that a separate analysis cannot be recommended (see also Doswell, C.A. III, 1987: The distinction between large-scale and mesoscale contribution to severe convection: A case study example. Wea. Forecasting, 2, 3-16).

In addition, it is confusing that all three rain events are supported by the dynamic factor, while in other places in the manuscript the opposite is written, such as in lines 119-122:

[Figure]

"The maximum rainfall was obtained in the rain shadow of the Negev Mountains, and the second most intense one was found in the Jordan valley, again, in the lee side of the Samarian Mountains. The dominance of convective over orographic elements suggests that sub-synoptic scale factors took place in this storm." These two statements can be confusing since it is not clear whether strong synoptic-scale forcing is important to these events or not.

Furthermore, the profile of one radiosonde is discussed (in 24-hour intervals). Unfortunately, the authors limit themselves to standard indices for analyzing the general thunderstorm potential. A discussion about whether the vertical profile supports the potential of heavy rain is not provided. In addition, the authors give no evidence to the hypothesis that the modified K-Index (MKI) gives better results in the east Mediterranean compared to standard indices (lines 321-324: "Despite the universality of stability indices developed to illustrate the potential for convection, few of them require adjustments and modifications to fit the area being analyzed. In this study, the modified KI version adopted for the eastern Mediterranean region, has shown to be a reliable predictor for convective rain centres and therefore a good precursor for floods." In this manuscript, the MKI just indicates the possibility of thunderstorms (as well as all other listed indices; lines 218-220 "The MKI distribution over the study region for the April 26, at 03, 09, 15 and 21 UTC, is shown in Figs. 7i-l, respectively. Values exceeding 25°C, indicating potential for thunderstorms, are co-located with the major rain centres at the hours 09, 15 and 21 UTC."). To convince the reader, a comparison with the K-Index can be useful.

A large part of the work is focused on the transport of moisture to the desert. The prevailing north-westerly flow and the transport of Mediterranean air are mentioned several times in the manuscript. However, based on WV satellite images it is also hypothesized that moisture from tropical regions had been advected, according to 315-317: "Moisture originating from tropical sources during such rainstorms enriches the mid-atmospheric levels, which makes the rain formation less sensitive to availability of

moisture in lower levels. Hence rain cells are not expected only over mountain up-slopes, but also over low terrains such as the one that caused the deadly flood in Tzafit creek." The authors may imply greater precipitation efficiency here, but this hypothesis is not further elaborated. Finally, in the conclusions, a hypothesis appears that is not given in the previous manuscript (lines 313-314: "Quasi-stationary upper level systems allow moisture accumulation causing the increase in precipitation amounts from one day to the next."). Without discussion, this hypothesis also remains without any evidence.

---

## Referee Comment (RC3) · Anonymous Referee #3 · 7 Jun 2020

Review of the manuscript "Atmospheric Conditions Leading to an Exceptional Fatal Flash Flood in the Negev Desert, Israel" by Uri Dayan, Itamar M. Lensky and Baruch Ziv

General

The paper presents a case study of an exceptional storm that not only had fatal impact, but was also a climatological outlier in the sense of its timing in the rain season, the distribution of precipitation relative to the regional orography, and the cyclone characteristics. The analysis involves a good combination of local observations and reanalysis data to infer the local instability conditions and the weather systems supporting the intense precipitation on the large-scale. Overall, the topic is important and the methodological approach is well designed. However, I have several reservations with regards

to the data, diagnostics and interpretation, as detailed below, requiring major revision of the manuscript before it can be considered for publication. The text is in many places too thin and not accurate enough, or backed by sufficient evidence, as I elaborate in the specific comments. Enhancing the introduction is necessary to place this case in a climatological context and provide more solid background about spring season rainfall in the region and Mediterranean cutoff lows.

Major comments

1. Introduction: this section is too thin to support the understanding of the unique aspects of this storm. In my view, more substantial background and recent literature should be included before the specific research aims are outlined. For example, I strongly recommend to include information on the following missing aspects: weather systems conductive for precipitation in the region in the transition seasons; what is the typical precipitation distribution in spring storms versus Cyprus lows; sharav cyclones; tropical systems affecting precipitation in the region; what are the typical precipitation intensities and how common is severe convection in such storms in this season; how common are cut-off lows?

2. The paragraph describing the aim of the study (L46-48) should be clarified. It is currently not clear what the authors mean by "one of the latest spring severe events..." does "latest" refer to the most recent one? Or to severe precipitation occurring very late in the rain season? Especially when the introduction is sufficiently expanded, it should be more clearly outlined what is unique about this storm/flood. For example, how well was it forecasted? What is unusual about the distribution of precipitation? What is unusual about this cut-off low? What else is unusual beyond its fatal impact? What do the authors mean by "its unique features" (L47)? The authors should avoid using such general terms and be more specific.

3. The construction of the reference list of cases is not outlined with sufficient details. Are there objective quantitative criteria? Lines 58-59 and L 66-67 are still too general.

Which streams are considered? What is the threshold for discharge? In how many stations? What is the reference region? Is the list restricted to cut-off lows? It will be good to reference Table 1 at this stage, and provide information on the precipitation in those reference storms, to then contrast the current storm in focus that produces a very different precipitation distribution.

4. There is a somewhat inconsistent usage of reanalysis datasets, with some fields taken from NCEP/NCAR, but PV and omega taken from ERA5. Why not analyse all fields from ERA5?

5. The motivation for examining and comparing the MCV values is not clearly revealed. Why not consider the shear vorticity as well and take the commonly-used relative vorticity as a measure for the intensity? Please justify this, especially given the fact that relative vorticity is anyway shown in Fig. 3. This clarification is again needed with regard to the list of reference cases. What do we learn from the high MCV? How do you interpret these differences?

Minor comments

1. L13: "one of the latest. . .3 decades" this is not clear and appears again throughout the manuscript. Please rephrase and clarify if you refer to the late timing in the season or to longer time scales.

2. L25: delete the mention of the temperature anomaly which is not shown, or add a section with this evidence to the results.

3. L43: what does "them" refer to?

4. L53: I suggest to replace "Material" by "Data and Methods"

5. L57: delete "to"

6. L58: what is meant by "maximum intensity"? it should be more accurate and clarify if it refers to precipitation/discharge/a vorticity measure/cyclone characteristic etc.

7. Fig. 1 caption: indicate the times in UTC for the date range; add "the red start marks the location of . . ."

8. Equations: replace the cross sign with a dot, to not confuse with vector notation.

9. L83: how is the depth of the cyclone estimated? Please provide the accurate measure.

10. L92: add "temperature" after "mean".

11. L94: remove one S from SSI.

12. L89-103: for each index, mention which values indicate severe convection or thunderstorms.

13. Eq 4: What is meant by RH850,700? why a modification of the K-index is needed for the eastern Mediterranean?

14. L104: replace "also used" by "analysed". Add "as" before "if". Is PW based on ERA5?

15. L116: "which activated convection" – this statement is not backed by evidence at this stage and should be deleted.

16. L118-119: the term "precipitative elements" is not a clear.

17. L121-122: the statement is again not backed by evidence at this stage, and it is not clear how this conclusion is reached, especially since it appears in the beginning of the results section. Furthermore, here and in L184-188 and throughout the text, the relationship between dynamical factors / orographic effects / convection / thermodynamic factors should be more clearly defined and distinguished from one another. For example, omega in ERA5 incorporates mass fluxes from convection, so its attribution as a clear dynamical diagnostic is not accurate. Please readdress these definitions, and outline them with regard to the analysis you carry out in this work.

18. L130-131: the transport of the dry air over the Levant is not consistent with the evolution of enhanced clouds at this stage.

19. Fig. 2: switch the locations of panels c and d; add "blue contours" after "m", and "black contours" after "hPa".

20. Fig. 3: replace "Course" by "Track"; add initials to the caption, e.g. "precipitable water (PW), CAPE (CA)"... ; replace here and throughout the manuscript (e.g., L200) "Km" with "km"; the blue text over the Med Sea is not visible; arrows in the late stages of the track are not visible;

21. Fig. 4 and accompanying text: it is unclear if this is PV or its anomaly (and how the anomaly is defined). I also recommend to switch the units to PVU and enlarge the domain.

22. L141-142: please add a reference to a climatology of such cutoff lows to demonstrate it is exceptional.

23. L165-166: This sentence is not clear. Can simplify by rewording to "... is expressed by enhanced easterly flow between the two vortices."

24. Fig. 5: In my view, the figure belongs more naturally in the discussion, and clearly after Fig. 6. In the figure, the term "blocking L" is confusing and should be reworded to "cut-off L".

25. Fig. 6: the arrows in (c) are not visible.

26. Fig. 7 caption: add in the end ", of 26 April 2018"; add units of MKI.

27. L 189: "cloud systems rotated cyclonically" – is there evidence for this advection as opposed to locally-produced clouds?

28. L187, 194: replace "ascendance" by "ascent".

29. L201: change 3c to 2c.

30. L202: add "reference" before "days"

31. Table 1: unclear how the depth is defined.

32. Section 3.3 and elsewhere: change "Kg" with "kg"

33. L226-227 "where it interacted with deep moist convection" this is a vague statement. Please clarify what you mean here.

34. L233: what is the evidence for "One is of tropical... at upper levels"?

35. L247: replace "one of the latest spring severe events" with "a severe storm occurring latest into spring".

36. L257-258: cutoff lows are not typical midlatitude cyclones, but rather a particular case in which the high-PV air is separated horizontally from the stratospheric reservoir in the upper troposphere.

37. L268-269: Please comment on the timing. This is occurring one day before the flood.

38. L299: "-5 K temperature anomaly". Please add more details on where is this anomaly located, at what vertical level, and add "not shown".

---

## Author Comment (AC1) · 5 Aug 2020

**1. General comments**

The study "Atmospheric Conditions Leading to an Exceptional Fatal Flash Flood in the Negev Desert, Israel" by Uri Dayan, Itamar M. Lensky and Baruch Ziv presents a thorough analysis of the synoptic and sub-synoptic patterns which produced a major storm in southern Israel, causing the death of 10 people. To my view, the significance of the paper is clear, given that there seems to be no sufficient knowledge to alert people ahead of such storms. Although analyzing one case study, the study has a substantial contribution to the understanding of flood-bearing rainstorms in Israel and its vicinity, as general conclusions are drawn and using a comparison to another eleven spring

storms that caused floods in the area. I find the analysis done along the cyclone track (MCV, speed) especially interesting, as it shows this cyclone intensified over Israel, and was therefore specifically intense over this region, and as it gives sort of reference values that can help in forecasting of severe storms. Furthermore, I think that the study of the atmospheric conditions during flood-bearing storms is much needed as these are the biggest natural hazard causing fatalities in the region. The scientific and presentation quality is good, although I have quite a few comments on both topics. Overall, I think that this study is interesting and important, and is worth publication in NHESS after addressing the points raised below. Three of my comments worth specific attention and are written under "specific comments". Other comments are smaller in nature (even if some are also conceptual) and most of them are technical in nature.

Response: We want to thank the reviewer for his/her valuable comments and considerable contribution for improving the quality of the research.

2. Specific comments

2.1 L160-174: These lines present an interesting idea; however, I am not sure the support given to it is enough. It is not clear to me if the slowing of the cyclone occurs every time there is a closed cyclone over the region or only when there is a dipole pattern.

Response: The present study analyzes the 25-26 April rainstorm as a case study, with a reference to previous storms in the same season (to which a sub section will be devoted in the results section), it does not aim to generalize the behavior of upper-level cyclones producing rain over the region. Thanks for the comment, we will make this point clear in the text.

2.2 If the dipole is needed, I am not sure whether the 500 hPa GPH anomaly can explain the dipole. The way I understand it from your description is that both the southern part of the high and the northern part of low induce easterly flow and slow the propagation of both systems. However, examining the actual synoptic maps (and not the

anomaly maps), I cannot see the highs. I can observe two ridges, one from to the west of the low and the other to the east of it (extending in Fig. 2c to the north of the cyclone). Do you account the ridge from the east for the slowing of the systems? I am not sure the pressure gradient is represented well enough by the anomaly maps.

Response: We agree that the credibility of the 500 hPa maps for reflecting the dipole is questionable. Alternatively, based on Hoskins et al. 1985 and Yamazaki and Itoh (2013) 'PV thinking', we superposed the conceptual diagram of a dipole type block (fig.5 in the original manuscript) on 3 300hPa PV maps every six hours. (see below Fig. 1).

This figure will replace the original Figs. 5 and 6, with the following explanation: The power of PV maps stems from the feature of PV anomalies (in the spatial sense) to induce circular flow out of their boundaries, cyclonic for positive and anticyclonic for negative. The induced easterlies, extending south of the negative anomaly (in the north) and these induced by the positive anomaly (to the south), slowed the propagation of both anomalies eastwards. These charts show a lobe of negative PV (blue) over the Balkans, north of the cutoff low, lingering over the Nile delta. The PV charts were extracted from the 0.25 deg resolution ERA5 reanalysis data (Copernicus Climate Change Service (C3S) 2017): ERA5: Fifth generation of ECMWF atmospheric reanalysis of the global climate. Copernicus Climate Change Service Climate Data Store (CDS), https://cds.climate.copernicus.eu/cdsapp#!/home).

2.3 L189-196: What is the timing of the rainfall that produced the flood? Rainfall is observed in the region of "Tzafit" both on 06-12UTC and in 12-18UTC, while significant omega is only seen in 15UTC. Please address this point, as it is crucial to understand whether ERA5 omega values are indeed accurate enough to capture such a small-scale rain spot.

Response: The rain that led to the Tzafit flood took place at April 26 and lasted ~1.5 hours, around 0930 UTC. Hence the Omega and MKI maps for 09 UTC (Figs 7b and

7j in the original version, respectively) intended to show the relevant mesoscale back-ground for the event. The blue color in the Omega map at the pertinent region indicates (Fig. 7b in the original version) upward motion in the order of 0.5 Pa/s, which cannot explain such an intense convection. We therefore shifted our focus to the instability (MKI) as the direct triggering factor, showing values (>30C), which are favorable for intense thunderstorms (Harats et al. 2010), the maximal for entire Israel. We modi-fied the text accordingly, omitted Omega from the new version of Fig. 7 (now Fig. 5) and replaced the six hours RADAR integration with one-hour integration, to fit satellite images, which were added to the figure (see below Fig. 2).

2.4 L225-236: I am wondering about the Tropical moisture source. Fig. 8 as well as Fig. 2c (and to some degree 2b) show southwesterly flow to Saudi Arabia, all consis-tent with your "tropical" source attribution. However, examining Fig 2a (and 2b) I can see convection triggered close to the cyclone's center, with northerly (Mediterranean) flow at low elevations. This makes me think the moisture source could have been Mediterranean to begin with, and this moisture (combined with the deep convection along the cyclone's track) could be the only moisture source to precipitation from this cyclone. The southwesterly flows in Saudi Arabia (Fig. 8 and Fig. 2c) existed probably at the night of the 25th of Apr, but to reach this region with sufficient moisture, you would probably had to look for southwesterly flows prior to this night, and this does not seems to be the case (Fig. 2a and b). Moreover, 18-24 h before arrival to "Tzafit Creek" the position of the red backtrajectory is in southern Syria. Going further back in time with the red trajectory gives Iraq (ground level) as the origination of the track. I am not familiar enough with this region, but I would expect it to be rather dry, and anyway, the track does not go all the way back to tropical latitudes. Please add some support for your moisture source determination, or correct this part of the text, as I suspect the Mediterranean could possibly be the only source in this case. One possible solution is to plot moisture content along the back-trajectory, and another one could be plotting a matrix of trajectories arriving to close-by locations.

Response: Following your comment, we further elaborated the possibility that tropical moisture contributed to the rain through highly detailed back-trajectories (see figure below for 26 April 09 UTC). Back-trajectories of 120 – h were derived using 50 km resolution ERA5 data, in 20 hPa interval from the surface up to 500 mb and are colored according to the specific humidity (g/kg). We now agree with you that the main moisture source was the Mediterranean. Additional moisture was transported in the mid-levels from Jordan through Syria, represented by the yellow band of trajectories in this attached figure and the red trajectory in Fig 9 (now 7). We modified the text accordingly (see below Fig. 3).

3. Technical (and more specific) comments

3.1 L10: Please state either that the ten casualties were among the people that die during the event (to the best of my knowledge, three more people died in other streams in Israel during the same event) or that among the flash floods one was specifically deadly, killing 10 people.

Response: We appreciate your attention very much as regarded to the number of casualties. Actually, 10 young people died in Tzafit in April 25, but another two people were killed in 25 April and one more in April 27, in spite of weakening of the rains during that day. We corrected the text accordingly.

3.2 L11-12: "The timing of the storm is also unique, at the end of the rainy season, when rain is relatively rare and spotty". Is it unusual to get extreme storms at the transitional seasons at the Negev desert? If not, please correct.

Response: The rainy season in the northern half of the Negev desert (Arad, Beer-Sheva, and Sde-Boker based on data from IMS for 1981-2010) are specified in the table below (mm). Source: Israeli Met. Service. Total May Apr Mar Feb Jan Dec Nov Oct Site/Month 132 1 6 20 28 31 23 13 10 Arad 190 4 4 29 40 48 38 18 9 Beer-Sheva 79 1 6 16 18 27 - 7 4 Sde-Boker During the late spring (Apr-May), the rainfall over the northern and central parts of the study area (5 - 10 mm) constitutes 4 - 9%

of the annual average. In spite of these negligible rain amounts, the number of flash flood events cannot be ignored. Based on 37 hydrometric stations operated by the Israeli Hydrological Service, covering the entire Negev desert, Kahana (1999) identified 59 "major floods". A "major floods" is a flood in which the recorded peak discharge reached the magnitude of a 5-year recurrence interval for the period 1947- 1994. Eight (14%) of these major floods occurred during April and May. The text will be modified accordingly. Ref: Kahana R. 1999: Synoptic Hydro-climatology of Major Floods in the Negev and Arava, Southern Israel. M.Sc Thesis, Institute of Earth Sciences, The Hebrew University of Jerusalem.

3.3 L13: Please rephrase "one of the latest spring severe events in the region during the last 3 decades" to "one of the most severe latest-spring events in the:" or something in that spirit.

Response: In the revised version we entitled the months April-May as "late spring". Accordingly, the phrase was changed to "one of the most severe late spring storms".

The same holds for L46-47. - L17: consider rephrasing "latest" to "last".

Response: Changed to "late", following our term "late spring".

3.4 L36-37: consider rephrasing "most weather-related fatal hazard" to "most fatal weather-related hazard".

Done.

3.5 L40-44: It is not clear whether most rainfall in the desert areas occurs during Dec-Feb or only over the non-desert parts of Israel. Please clarify this point.

Response: Most of the rainfall occurs in DJF all over Israel and the northern half of the Negev desert, as can be inferred from the table in our response to comment #2 above. The relevant text now reads: "... majority of the annual precipitation in Israel, excluding its southern most 100 kilometers, is associated with Mediterranean cyclones . . ... two-thirds of which occur during December through February".

3.6 L55: "over which floods took place" during this event? In general? Please explain.

Response: The phrase "over which floods took place" was changed to "in which the Tzafit flood took place".

3.7 L58: Please explain how "maximum intensity" is defined.

Response: After further consultation, we arrived at the conclusion that from a meteorological point of view, no distinct difference can be noted in the rain intensity between April 25 and 26. The relevant notions were removed, and the text modifications are: In L58: The phrase "when it reached its maximum intensity" was replaced by "when the Tzafit flood took place". In L117: The phrase "reached its maximum intensity at April 26" was removed.

3.8 L58: Can you please elaborate on the 11 storms? Are these the only storms during Apr-May over this period? Are these the highest discharge storms (as suggested in L66)? Are these the same storms listed in Table 1? If these are the same storms, please number the different storms in the list, as it is hard to find 11 storms in it. If they are not the same storms, please list these storms in a table as well.

Response: The storms referred to in line 58 are the same as these mentioned in line 66. To avoid ambiguity the phrase "to other 11 storms, spread over 28 days, in which discharge was observed over the study area, during April and May for the years 1986 – 2018, entitled hereafter the 'reference period'" was changed to: "to a set of 11 storms (including the one studied here), spread over 30 days, in which discharge was observed over at least one of the 37 hydrometric stations of the Israeli Hydrological Service that has been operative during 1986 – 2018 over the study area. These storms, which occurred during April and May, are hereafter entitled as the 'reference storms'". We listed the storms as requested and changed the text accordingly. In addition, we now devote a subsection in the "results" section for a brief description of the remaining 10 storms.

3.9 L61: what is the spatiotemporal resolution of the radar data used? If I am not mistaken, Marra & Morin (2015) data come from another radar in the region.

Response: The spatial resolution of the radar is 250 m in the radial direction and 1 in the azimuthal. This data is rectified to $1 \times 1$ Km grid. The radar temporal resolution is 5 min, but in the images shown in Fig. 6 (new version) the data are integrated over 1 hour. The radar used belongs to the Israel Met. Service, located in Beit Dagan. This is added to the text.

3.10 Figure 1: It is not clear whether this map comes from accumulated IMS radar data ("RADAR" is stated in the figure's caption) or from the "rain measuring network" (L60). The shape of the contours looks as if coming from an interpolation of gauges. Please explain and change the text referring the figure or the figure's caption accordingly. It seems radar data is more appropriate to use since rain gauges in the analyzed region seems scarce.

Response: The RADAR images indeed incorporate rain gauges measurements, based on the integration method identical to that used by the INCA system (Haiden et al. 2011). We added this to the text (L61-62) and to the captions of Figs. 1 and 7 (now 6). We also added the appropriate reference: Haiden, T., A. Kann, C. Wittmann, G. Pistotnik, B. Bica, and C. Gruber, 2011: The Integrated Nowcasting through Comprehensive Analysis (INCA) System and Its Validation over the Eastern Alpine Region. Wea. Forecasting, 26, 166–183, https://doi.org/10.1175/2010WAF2222451.1.

3.11 L66: What do you mean in "the data of floods"? Does this refer to hydrographs? Peak discharge?

Response: The flood data for the reference storms is the discharge based on 37 hydrometric stations that were operated by the Israeli Hydrological Service covering the entire Negev desert (see our response to comment 3.8).

3.12 L70: please add a reference to this statement. Consider one (or more) of the

following for this specific region:

Response: We modified the relevant text and added refs, for the general context, and specific region, as follows: "In general, the atmospheric processes responsible for flash flood-producing rainstorms act in concert at the synoptic and sub-synoptic scales (Bardossy and Filiz, 2005). Severe convection during the spring over the Negev desert region are often created as a combination of sub-synoptic processes imbedded in synoptic systems that may affect this region (Dayan and Morin, 2006; Armon et al.,2018)." Refs: Armon, M. Morin, E., and Enzel,Y.: Overview of modern atmospheric patterns controlling rainfall and floods into the Dead Sea: Implications for the lake's sedimentology and paleohydrology, QUATERNARY SCIENCE REVIEWS, Volume: 216, Pages: 58-73 DOI: 0.1016/j.quascirev.2019.06.005, 2019. Bardossy, A., and Filiz, F.: Identification of flood producing atmospheric circulation patterns, J. of Hydrology, Volume 313, Issues 1–2, 5 November 2005, Pages 48-57, https://doi.org/10.1016/j.jhydrol.2005.02.006. Dayan, U., and Morin, E.: "Flash flood–producing rainstorms over the Dead Sea: A review." New frontiers in Dead Sea paleoenvironmental research 401 (2006): 53.

3.13 L69-74: please explain why you use different datasets (ERA5 and NCEP/NCAR). Do coarse scale ERA5 data the same (or very similar) to NCEP/NCAR data? If not, the use of ERA5 data both in the synoptic scale and in the meso-scale is probably better.

Response: The reason for using different data sources is availability. Unfortunately, we have no access to ECMWF data of 2.5×2.5 deg on the one hand, and there is no fine resolution NCEP data of 0.25×0.25 deg on the other. However, a comparison of several parallel maps from the two data bases used shows a satisfying fit (see example of SLP below). The latter is mentioned now in the text. (see below - Fig 4.)

3.14 L83: What are the units of the cyclone depth? is it m or hPa? Do you measure the depth at the 500 hPa as well? Does V refer to the 500 hPa?

Response: The cyclones' depth was derived from the 500 hPa GPH and are expressed

in m. In the text, before "The tangential..." we inserted "The cyclone depth (in meters) is the difference between the central height and that of the outer most closed isohypse".

3.15 L90: "usually 500 hPa" – is this the case here or do you consider another pressure level?

Response: The text has been changed to "Lifted Index (LI, Galway 1956): Temperature difference between the environment and an air parcel lifted adiabatically from 2 m above the surface to 500 hPa. Negative values indicate instability."

3.16 L98-103: did you consider using the Beit Dagan sounding data for the MKI index as well? This could help understanding whether there are significant differences in ERA5 data in respect to the sounding data, which is not situated within your study area.

Response: The MKI calculated from Beit Dagan sounding for 26 April 12 UTC was 13.2 deg, lower than that implied by the map (Fig. 7j, now 6j, which correspond to 09 UTC). However, the values in this map for Tzafit creek was 10 deg higher (>30 deg). This difference can be attributed to the warming of the air while moving inland about 100 Km. The correspondence between the maxima in MKI and the major rain cells is now incorporated in our explanation for the timing and location of this rain event, and the other one, in the Jordan Valley. Thank you for drawing our attention to this point.

3.17 L104-105: Where does PW come from? Is this based on ERA5? Please indicate it in the text.

Response: The PW was taken from the soundings of Beit Dagan. We now state it in the text.

3.18 L107-109: Can you describe what kind of back-trajectories did you used? Which atmospheric model was used (NCEP/NCAR reanalysis / ERA5 / Other data)? What kind of levels did you use (model / pressure / isentropic)?

Response: The derivation of the trajectories shown in of Fig. 9 (now 7) is based on

the Global Data Assimilation System (GDAS 1° by 1°), which is used by the National Center for Environmental Prediction (NCEP) Global Forecast System (GFS) model to place observations into a gridded model space for initializing weather forecasts with observed data. The vertical motion used is the "Model vertical velocity" that is derived from the gridded data. The level used is pressure level. This is now added to the text.

3.19 Fig. 2: Could you please describe the MODIS imagery? Does the image come from 12UTC as well? Is it from a specific MODIS (Terra/Aqua)? If so, please write it.

Response: We apologize for missing this information as regarded to the satellite used: the satellite imagery in Fig. 2 were retrieved from the NASA Worldview (https://worldview.earthdata.nasa.gov/). The satellite, operated by NOAA, is the Suomi National Polar-orbiting Partnership (Suomi NPP), a polar orbiting weather satellite carrying, among other instruments, a Visible/Infrared Imager Radiometer. The satellite imageries displayed are along the following NPP passes: April 24 (Fig. 2a) – 11:50 UTC, 25 April (Fig.2b) – 11:30 UTC, 26 April (Fig. 2c) – 11:15 UTC, and 27 April (Fig. 2d) – 10:55 UTC. The missing information is added to the caption of Fig. 2.

3.20 L131- 149 and Fig. 3: It is not clear to me how did you track the cyclone at the 500 hPa. Was it through the large scale NCEP/NCAR data or the ERA5? Or possibly using satellite images? Please explain it either in the materials or specifically when referring to the cyclone's movement.

Response: The 500 hPa cyclone center was tracked according to the large scale (2.5×2.5 deg) NCEP/NCAR data in a 12-hours increment. This notion is now added to the caption of Fig. 3.

3.21 Fig. 4: Please correct the caption ("with maximum of K: : :"), consider choosing a more appropriate color scale (centered at zero or starting at zero), a better in-figure-caption units (currently it is "10Ё Ȩ1 K m**2 kg**-1 s**-1"), and possibly use PVU units instead.

Response: We updated the figure as suggested (see below last figure - Fig 5).

3.22 L141: "This is rather exceptional: : :" – did you compare the value to other values from your reference period? This could be interesting and could also put the results in a broader context.

Response: In the new version, the April 26, 2018 storm is compared to the other 10 reference storms in a special sub-section in the results section (see also our response to comment 3.8). It is shown that in terms of curvature vorticity, approximated by the MCV, this storm is the most intense (see Table 1). However, we now refrain from giving the storm any title, except for a brief description of some quantitative aspects of its weather phenomena in the introduction.

3.23 L185 please replace the square sign with a degree sign.

Done.

3.24 Figure 7, middle panels: Do these panels represent radar, gauges or a combination of radar and gauges? Panel f seems to be some combination of gauges and radar, but this is not clear from the caption or "materials" section.

Response: The RADAR images in this figure incorporate rain gauges measurements, based on the integration method identical to that used by the INCA system (Haiden et al. 2011), as in Fig. 1 (see our response to comment 3.10). We added this to the figures' captions.

3.25 L181: what do you mean by "integrated"? Does this refers to the combination of sources or to the accumulation of rainfall? Please add units to the MKI legend or caption.

Response: The radar data and the rainfall measurements, on which the "radar" images in Fig. 7 (now 6) are based, were temporally integrated over 6 hours each in the original figure, and now are integrated over one hour (will be added to the text, see also our response to comment 2.3). The MKI units are degrees. It is added to the figure caption.

3.26 L194: "-10 Pa s-1" do you mean -1 Pa s-1? If not, please correct Fig. 7's legend.

Response: Pascal s-1 is correct. Unfortunately, we found a factor 10 error in line 187 and corrected "10" to "1".

3.27 L244: Please see previous comment about the "unique timing" of the storm.

Response: Please see our response to comment 2.3.

3.28 L245: This paragraph seems to me as a summary paragraph, however the term "Syrian low" was only introduced in L245. Could you please either remove this line from the summary or describe this term in the introduction/results sections?

Response: Thank you for the suggestion. The 'Syrian Low' is now presented and explained in the introduction, following yours and the comments of the other reviewers.

3.29 L285: Cape values in Fig. 2 reach to 909 and not to 1000. Please correct or explain this notion.

Response: The phrase "values reached 1000 J K g-1" is now replaced by "values exceeded 900 J K g-1."

3.30 L286-289: Contribution of tropical moisture was shown in a number of studies; however, I am not sure this is the case here. Please address the previous comments about this subject. If you are certain about the tropical origins, please add more references or add "e.g.". –

Response: We agree with this comment, as is reflected from our response to comment 2.4.

3.31 L290-294: The point raised here is interesting and fits well a discussion section. Since this section of the paper is written in bullets, it looks like a summary or a conclusion. Please give a better distinction between the summary and the discussion. This will help readers to understand which parts in this section should only be treated as a discussion and are therefore introduced firstly in this section, and which are summarizing prior sections. Perhaps the discussion can remain as one section and the summary can go with the conclusions, and in this way you will reduce redundancy.

Response: Following your suggestion we separate the last section to "discussion" and then - "summary".

3.32 L299: Please either add "not shown" after the "-5 K" or show this anomaly. Currently, there is an impression this anomaly is given in Fig 6.

Response: We modified the phrase "a -5 K temperature anomaly over the region" to "a -5 K temperature anomaly in the 500 hPa level over the Levant (not shown)".

3.33 L303-L310 "unique intensity", "exceptionally severe rainstorm", "severity of this rainstorm": what exactly all of these expressions refer to? Do you mean the MCV? The rainfall (is it really that rare)? The casualties (were they influenced by the extreme severity, bad luck, or bad circumstances)? Please explain this point, and better to do it in previous sections of the paper.

Response: We now briefly describe some quantitative aspects of the weather phenomena of the storm in the introduction, and refrain from giving it any title, such as "exceptional", "unique" (see also our response to comment 3.22).

3.34 L313-314: Is this the case here? Was it written previously?

Response: We omitted this point.

3.35 L315-317: Please see previous comments about tropical moisture origins (although this conclusion seems reasonable anyway).

Response: Due to the omission of the tropics as a relevant moisture source, the sentence is modified to: "The average distribution of rain associated with Syrian low is shown by Saaroni et al. (2010) under the title "deep low to the east". This distribution shows maximum rain in the coastal plain and along the mountain ridges facing the north westerly winds, i.e., orographic. The distribution of the rain in the present storm

has maxima over low terrains, such as the one that caused the fatal flash flood in Tzafit creek. The explanation for that may be the instability that dominated the event and intrusion of moisture in the mid-level that could skip over the mountain ridges."

3.36 L318-320: Please also see previous comment regarding L189-196.

Response: As we responded to comment 2.3, the new version of Fig. 7 (now Fig. 5) does not contain Omega maps anymore. Moreover, the Omega field is not referred to as a basis for explaining the development of major rain cells.

———————————————————

[Figure]

[Figure]

**Fig. 1.** PV distribution on 300 hPa, starting from April 25, 2018, 12 UTC, in 6-hour intervals, covering Europe and the Mediterranean Basin, on which a conceptual diagram of a dipole type block (following Yama

**Fig. 2.** Fig. 5. Set of maps for April 26 2018, 09 and 12 UTC: Satellite image of MSG ch9 (10.8 um) (a,b); Radar imagery of one hour integrated rain depth (mm), (c,d); Relative vorticity and wind (e,f) and MKI

**Fig. 3.** Air back-trajectories arriving at Tzafit (31.1N, 35.2E) in 26 April 2018 09 UTC. The colors denote specific humidity along the trajectory (g/Kg).

[Figure]

**Fig. 4.** example of SLP

PVU ($10^{-6}$ K m$^{-2}$ kg$^{-1}$ s$^{-1}$)

**Fig. 5.** Figure 4

---

## Author Comment (AC2) · 6 Aug 2020

This manuscript deals with a very interesting case of a high-impact storm. There were floods in the Negev desert in southern Israel, in which unfortunately 10 people died. The event had not been expected as it hit at the end of the cold season. A scientific investigation of this storm is important in order to understand the underlying processes and to improve the forecast for such flooding events. Additionally, a climatological classification as carried out in the work, helps to better estimate the potential of these events. However, the work is essentially limited to large-scale processes that are also not completely discussed. In addition, there are attempts to connect small-scale processes to the synoptic scale, which fail because of the separation between dynamics and thermodynamics and not least because of the selection of data and the way these

are presented. Finally, the manuscript gives repeatedly undisputed hypotheses, as well as some inconsistencies. The main goal of this work as given in lines 50-52 ("In section 3 we describe the event and identify the unique dynamic and thermodynamic conditions that lead to the severe convection, as well as the sources of moisture for the rain formation in this storm.") needs to be elaborated more, especially with respect to across-scale processes with respect to rain production. Evidence needs to be given to the hypotheses that are presented.

Response: Following your valuable comments, we intend to rewrite the discussion section.

Specific comments:

1) The manuscript is focused on the analysis of the large-scale flow, which is compared with similar flooding events. The intensity and track of the corresponding cut-off low in 500 hPa is unusual for the season. Parts of the work, however, disagree on whether the event is a typical weather situation, with flooding occurring outside the rainy season in the desert or whether it is a unique situation (compare lines 42-45: "The rest of the annual rainfall occurs in the transitional seasons and is contributed by precipitating tropical synoptic-scale systems and by Cyprus Lows. A significant part of them occur in the desert areas and are characterized as intense rain events of small spatial extent and short duration, some of which produce flash floods (Kahana et al. 2002; Dayan and Morin 2006, Greenbaum et al. 2010)." and lines 245-247: "The location of the surface cyclone was similar to the 'Syrian low', defined by Kahana et al. (2002) as one of the major systems causing floods in the Negev Desert." with line 51: "unique dynamic and thermodynamic conditions"). The impression is that the large-scale weather conditions are well known for flooding, whereas only the temporal appearance and the intensity and coverage were unique. Readers unfamiliar with the given weather patterns, i.e. 'Syrian low' and 'Cyprus Lows', can be confused which of these two is mainly associated with flooding events. It would be good to explain these weather patterns at the beginning of the text in some detail, e.g. mid-level flow and surface pressure field.

[Figure]

Response: We agree that the status of this storm, in the regional and seasonal context, is not set properly, and leave the reader with an unclear impression. We will include (a) climatological background and (b) a general statement in the introduction section:

(a) "The majority of the annual precipitation in Israel is associated with Mediterranean cyclones, in the stage when they reach the Middle-East (i.e., Cyprus Lows, HMSO 1962; Saaroni et al. 2010; Zappa et al. 2015). Two-thirds of the rainfall occur during December through February (Alpert et al. 2004; Ziv et al. 2006). The focus in this study is on the Negev desert and the Judean desert (Fig. 1), hereafter the 'study region'. The climatic regimes of the study region span from semiarid in the north to an arid in the center and the south (south of 3115', Ziv et al 2014). The rainfall during the late spring months, Apr-May, contribute 5 - 10 mm (4 - 9%) of the annual average over the northern and central parts of the Negev desert. In spite of these negligible rain amounts, the number of flash flood events in late spring cannot be ignored. The flood regime in the study region was analyzed by Kahana (1999), based on 37 hydrometric stations operated by the Israeli Hydrological Service. He identified 59 "major floods", i.e. floods in which the recorded peak discharge reached the magnitude of a 5-year recurrence interval, for the period 1947- 1994 at least in one watershed. Eight (14%) of these major floods occurred during the late spring. The main source of major floods in the late spring over the study region is the "Active Red Sea Trough" (ARST, Kahana et al., 2002). The ARST is most frequent during fall and spring (Sharon, 1978; Sharon and Kutiel, 1986; Dayan et al., 2001). This is a lower-level trough extending from equatorial eastern Africa into the eastern and southern Israel along the Red Sea and is accompanied by a pronounced upper-level trough over Egypt. At times, it initiates severe convective storms. The secondary source of major floods are Cyprus Lows that cross the eastern coast of the Mediterranean, but remain intense. Kahana et al, (2002) entitled them "Syrian Lows". Under the influence of Syrian lows, the Levant is subjected to surface north-westerly flow, enriched with moisture from the Mediterranean. While flowing onshore, rains are produced over the northern Negev and southern Judean desert and, due to orography, over the north and western slopes of the Negev and

Judean Mountains.

(b) The storm analyzed here was severe in several aspects. One is the number of fatalities, 13, which is a record breaking for Israel. Second, a part of the Zin basin (draining the center of the Negev desert) experienced precipitation intensities reaching 75-100-year return period, resulting in discharge magnitudes of 10-50-year return period (Rinat et al. 2020). The third aspect is the rainfall totals for the storm that reached 40-50 mm over large parts of the study region, i.e., 10 times the monthly long-term mean (IMS, 2018). The aim of this study is to assess the severity of this storm in the seasonal perspective and to analyze the atmospheric conditions that explain it."

Reference: Rinat, Y., Marra, F., Armon, M., Metzger, A., Levi, Y., Khain, P., Vadislavsky, E., Rosensaft, M. and Morin E. Hydrometeorological analysis and forecasting of a 3-day flash-flood triggering desert rainstorm, NHESS doi.org/10.5194/nhess-2020-189, 2020 Moreover, this storm is compared to the other 10 "reference storms" that occurred in the same months during the latest 33 years, now in the last subsection of Sec. 3. In the summary the status of this storm is addressed as well.

2) Apart from this analysis of mid-level and SFC flow, hypotheses are (repeatedly) raised that are not discussed further: Lines 311-312 "The combination of a cut-off low with small radius and large hypsometric depth, implies high curvature relative vorticity, with strong dynamical forcing on rain formation." At this point, the authors need to be more precise. "Strong dynamical forcing" refers to quasi-geostrophic processes, and large-scale lift is the order of cm/h. Does QG lift affect rain formation directly? Moreover, only the contribution by differential cyclonic curvature vorticity advection to QG lift is mentioned. Since the weather charts indicate north-westerly flow, the reader may wonder if a cold air advection maximum may cancel QG lift in a similar event.

Response: We accept this critical comment concerning our considering the synoptic (i.e., the large dynamic processes) and thermodynamic elements (i.e., mesoscale processes permitting deep convection) separately. Therefore, we intend to reorganize the

manuscript in the various relevant parts, in particular the discussion section, considering the synoptic-scale as the background for the smaller-scales, which characterized the major rain cells, as follows:

Following the quasi-geostrophic approach, most convective outbreaks occur in broad southwesterly flow aloft ahead of an approaching trough (Doswell, 1987). In April 26, under the northwesterly flow behind the upper-level cyclonic system, its direct dynamic supporting effect on rain formation is expected to be weak, or even negative (Fig. 2c). At this sector, negative upper-level vorticity advection and lower-level cold advection are typical, and both induce subsidence in the mid-levels. Inspection of the lower-level temperature and wind fields (not shown in the paper, shown for you) indicate that at that stage, the cold core associated with the upper-level cyclone was centered over the Negev desert. The lower-level winds over the study region were parallel to the isotherms, so that no temperature advection existed. Similar configuration existed with respect to the upper-level vorticity field, implying that vorticity advection also did not take place there. (See Fig. 1 below)

Moreover, the omega field at the mid-levels in the synoptic scale over the region was near zero. The above implies that the synoptic scale dynamics did not have a direct effect on the rain formation on that day. The major synoptic factor that directly contributed to the rain formation in this storm is the wind. One implication is the onshore moisture transport, accompanied by an uplift imparted by its encounter with the coastline and later on, with the mountain ridges. The other is the upper-level cold advection, leading to thermal instability. Actually, only a small fraction of the rainfall in this storm was orographic (over Judean Mountains), whereas the majority of the rain was observed far from the coastline and beyond the water divide of the mountain ridges (Fig. 1, note the divide line). The major rain cells were convective: the one that produced the flood in Tzafit, over 60 km inland, and the one developed near Beit Shean, in the Jordan valley, >200 m BSL. The dominance of convective over orographic rain suggests that instability was the prominent factor in this storm. In the EM, rains associated

with Mediterranean cyclones, as in the case studied here, are convective in nature. In the winter season, cold air originating from south Europe moves over the EM, interacts with the underneath warmer Mediterranean water and enters Israel (Shay-El and Alpert 1991, Saaroni et al. 2010). In the late spring, Europe becomes warmer, whereas the Mediterranean remains cool, due to its lagged response to the annual cycle, the passage of a Cyprus low over the EM does not necessarily lead to instability. In the case studied here, the instability can be attributed to a negative temperature anomaly in the upper-levels (in the order of -5 K in 500 hPa, not shown) that covered the southern Levant, as a part of the cutoff low. Beit Dagan radiosonde profile for April 26, 12 UTC indicates that an air-parcel had to be lifted up to 1 km to trigger convection. Indeed, the negative Omega values during the storm were in the order of 1Pa s-1, implies that an entire day would be needed to reach the level of free convection. Furthermore, we observed that the major rain cells were in the order of tens of kilometers, implying that they experienced forcing in the meso-scale rather than in the synoptic. This is consistent with Doswell (1987), who showed how the thermodynamic environment favorable for intense convection is formed through large scale processes. Concerning the small radius and large hypsometric depth of the cutoff lows, we referred to the amplitude of the upper-level cyclone as a coarse measure for the general strength of the storm.

3) And what about shear vorticity?

Response: The 'measure of the curvature vorticity' (MCV) is a proxy for average curvature vorticity over the upper-level cyclone, so it evaluates its overall intensity. Since this cyclone moved along the latitudes of Israel, while the Subtropical jet shifted >7 deg southward, the jet's associated wind shear vorticity was regarded as marginal, so that the MCV can be considered as an estimate for the full vorticity. We will add this notion in the "methodology" section before introducing the MCV. (See Fig. 2 below).

4) Lines 193-195: "It should be noted that the three rain centers are located within a region of negative Omega (ascendance, Fig. 7c), with an extremum value of -10 Pa s-1. This implies that these rain systems are dynamically supported." In this manuscript,

the "dynamic factor" (quasi-geostrophic lift) is analyzed separately from the "thermody-namic factor" (thunderstorm) (lines 123-125: "Two complementing factors contributed to the rain formation. One is dynamic, i.e., vertical ascent, associated with the cyclonic system described below (Sec. 3.2). The other is thermodynamic, which is composed of instability and moisture supply, described in Sec. 3.3.").

Response: We accept and adopt the suggested approach concerning the separation between the "dynamic" and the "thermodynamic" factors. Also, we found an error in our text. The values of OMEGA reflect even lower forcing of 1 Pa/s, implying a weaker meso-scale lifting.

5) There are two criticisms to approach. 5.1) First, the omega fields show a combina-tion of lift on different scales, including convection parametrization. It is therefore not possible to conclude that the given omega fields indicate just dynamic lift.

Response: Following your argument, and due to the small amplitude found in Omega field even in the meso-scale resolution (Fig. 7), we will exclude the Omega maps from the paper. The figure below is planned to replace Fig. 7. (See Fig. 3 below).

5.2) Secondly, both factors influence each other, so that a separate analysis cannot be recommended (see also Doswell, C.A. III, 1987: The distinction between large-scale and mesoscale contribution to severe convection: A case study example. Wea. Forecasting, 2, 3-16).

Response: We accept this comment as addressed in our response to comment #2.

5.3) In addition, it is confusing that all three rain events are supported by the dynamic factor, while in other places in the manuscript the opposite is written, such as in lines 119-122: "The maximum rainfall was obtained in the rain shadow of the Negev Moun-tains, and the second most intense one was found in the Jordan valley, again, in the lee side of the Samarian Mountains. The dominance of convective over orographic elements suggests that sub-synoptic scale factors took place in this storm." These two

statements can be confusing since it is not clear whether strong synoptic-scale forcing is important to these events or not.

Response: Thank you for this comment, the formation of major rain cells cannot be attributed to uplift induced by upper-level dynamics as stressed in our response to comment #2. The first major rain cell that developed over Tzafit indeed was form where the terrain has a negative (though moderate) slope with respect to the northwestern wind there. As for the major rain cell developed near Beit Shean, the winds there were northerly, so that it was not formed at the rain shadow of the Samarian Mountains (see Fig. 1). We intend to propose that horizontal confluence was exerted on the surface flow by the conical shape of the Jordan Valley there. We accept that instability is the major factor, and even support it by MKI maps (see Fig. 3 below).

6) Furthermore, the profile of one radiosonde is discussed (in 24-hour intervals). Unfortunately, the authors limit themselves to standard indices for analyzing the general thunderstorm potential. A discussion about whether the vertical profile supports the potential of heavy rain is not provided.

Response: Unfortunately, only one sounding station is operative in Israel (Beit Dagan), and even this one has only 2 observations a day (00 and 12 UTC) and is located dozens of kilometers from the major rain cells. However, analysis of the thermodynamic diagram (i.e. Tephigram) for April 26, 12 UTC indicates that in spite of a shallow stable layer that was found around 700 hPa level, a convective cloud could develop if an air parcel had been lifted to 900 hPa level, with top reaching 330 hPa level. The higher instability observed further inland (see MKI map in our response to comment #6) explain why the main convective activity was observed further offshore. Moreover, the locality of the rain cells typifying this storm demonstrates the minor importance of the direct dynamic synoptic forcing relative to that of instability. This will be stressed in the discussion section of the revised manuscript.

7) In addition, the authors give no evidence to the hypothesis that the modified K-

Index (MKI) gives better results in the east Mediterranean compared to standard indices (lines 321-324: "Despite the universality of stability indices developed to illustrate the potential for convection, few of them require adjustments and modifications to fit the area being analyzed. In this study, the modified KI version adopted for the eastern Mediterranean region, has shown to be a reliable predictor for convective rain centres and therefore a good precursor for floods." In this manuscript, the MKI just indicates the possibility of thunderstorms (as well as all other listed indices; lines 218-220 "The MKI distribution over the study region for the April 26, at 03, 09, 15 and 21 UTC, is shown in Figs. 7i-l, respectively. Values exceeding 25C, indicating potential for thunderstorms, are co-located with the major rain centres at the hours 09, 15 and 21 UTC."). To convince the reader, a comparison with the K-Index can be useful.

Response: The modification of KI, to MKI, are defined and referenced in the methodology section. The MKI was proposed and examined by Harats et al. (2010), following cases of false thunderstorm alarms in the Mediterranean region. The MKI differs from the KI in that the 1st term, the temperature difference between 850 and 500 hPa, is multiplied by the average relative humidity over the 850 and 700 levels. Hence, whenever the lower- or mid-levels are dry, the MKI is reduced compared to the KI. The MKI was further elaborated and tested on a large number of rain cells over the Mediterranean Basin by Ziv et al. (2016). This is demonstrated by comparing KI and MKI for April 26, 07 and 09 UTC. The main difference is in the high KI values compared to the MKI over arid regions. (See Fig. 4 below).

Ref: Ziv, B., Harats, N., Morin, E. et al. Can severe rain events over the Mediterranean region be detected through simple numerical indices? Nat Hazards 83, 1197–1212 (2016). https://doi.org/10.1007/s11069-016-2385-y.

8) A large part of the work is focused on the transport of moisture to the desert. The prevailing north-westerly flow and the transport of Mediterranean air are mentioned several times in the manuscript. However, based on WV satellite images it is also hypothesized that moisture from tropical regions had been advected, according to 315-

317: "Moisture originating from tropical sources during such rainstorms enriches the mid-atmospheric levels, which makes the rain formation less sensitive to availability of moisture in lower levels. Hence rain cells are not expected only over mountain upslopes, but also over low terrains such as the one that caused the deadly flood in Tzafit creek." The authors may imply greater precipitation efficiency here, but this hypothesis is not further elaborated.

Response: Following this comment, we further elaborated the possibility that tropical moisture contributed to the rain through highly detailed back-trajectories (See Fig. 5 below for April 26, 09 UTC). Back-trajectories of 120 – h were derived using a 50 km resolution ERA5 data, in 20 hPa interval from the surface up to 500 mb and are colored according to the specific humidity (g/kg). We now agree that the main moisture source was the Mediterranean. Additional moisture was transported in the mid-levels from Jordan through Syria, as represented by the yellow band of trajectories in this attached figure and the red trajectory in Fig 9. We intend to modify the text accordingly. Concerning the possibility of regions that not exposed to the direct moisture advection from the Mediterranean, this can be explained as follows: The lower-level system in the storm analyzed here, can be considered as a 'Syrian low' (Kahana et al., 2002), which belongs to the Mediterranean cyclones. The Syrian low resembles the 'deep low to the east' as one of the 7 types of Cyprus low defined by Alpert et al. (2004). Saaroni et al. (2010) showed that most of the rain associated with the 'deep low to the east' is distributed over the coastal regions and the western slopes of the Judean Mountains, facing the offshore northwesterly winds from the Mediterranean. This indicates that the Judean Mountains, extending up to 800-1000 m, block effectively this moisture, which is presumably, concentrated in the lower-levels. The heavy rains that were observed inland in this storm and the signature of non-orographic major rain cells in the rain maps indicate that the moisture sources were not limited to the lower-levels. The presence of moisture at the mid-levels can be deduced from back-trajectory arriving at Tzafit at the time where the flood occurred (Fig. 9 and the figure shown above), showing band of mid-level moist air that originated east of the Levant, revolved around the cyclone

center and entered the region from the west.

9) Finally, in the conclusions, a hypothesis appears that is not given in the previous manuscript (lines 313-314: "Quasi-stationary upper level systems allow moisture accumulation causing the increase in precipitation amounts from one day to the next."). Without discussion, this hypothesis also remains without any evidence.

Response: This conclusion is based, in addition to the present case, on a case of an Active RST (Ziv et al. 2005), which was stationary in the lower-levels for 3 days, during which its activity increased consistently. Since 2 cases are far from being a basis for such a conclusion, we will omit it. Ref: Ziv B., U. Dayan and D. Sharon, 2005: A mid-winter, tropical extreme flood-producing storm in southern Israel: Synoptic scale analysis, Meteor. Atmos. Phys., 88(1-2): 53–63.
* * *
[Figure]

**Fig. 1.** Upper-level vorticity advection and lower-level temperature advection, based on the NCEP reanalysis data, with 2.5×2.5 deg resolution.

[Figure]

**Fig. 2.** The zonal component of the wind averaged over 25-27 April 2018 in 500 (left) and 300 (right) hPa

**Fig. 3.** Set of maps for April 26 2018, 09 and 12 UTC: Satellite image of MSG ch9 (10.8 ïA■m) (a,b); Radar imagery of one hour integrated rain depth (mm), (c,d); Relative vorticity and wind (e,f) and MKI (g,h).

**MKI**

**KI**

Fig. 4. MKI vs. KI

**Fig. 5.** Air back-trajectories arriving at Tzafit (31.1N, 35.2E) in April 26, 2018 09 UTC. The colors denote specific humidity along the trajectory (g/Kg).

---

## Author Comment (AC3) · 6 Aug 2020

General

The paper presents a case study of an exceptional storm that not only had fatal impact, but was also a climatological outlier in the sense of its timing in the rain season, the distribution of precipitation relative to the regional orography, and the cyclone characteristics. The analysis involves a good combination of local observations and reanalysis data to infer the local instability conditions and the weather systems supporting the intense precipitation on the large-scale. Overall, the topic is important and the methodological approach is well designed. However, I have several reservations with regards to the data, diagnostics and interpretation, as detailed below, requiring major revision

of the manuscript before it can be considered for publication. The text is in many places too thin and not accurate enough, or backed by sufficient evidence, as I elaborate in the specific comments. Enhancing the introduction is necessary to place this case in a climatological context and provide more solid background about spring season rainfall in the region and Mediterranean cutoff lows. We wish to thank the reviewer for his/her constructive comments and considerable contribution, which we believe will improve the paper.

Major comments

1. Introduction: this section is too thin to support the understanding of the unique aspects of this storm. In my view, more substantial background and recent literature should be included before the specific research aims are outlined. For example, I strongly recommend to include information on the following missing aspects: weather systems conductive for precipitation in the region in the transition seasons; what is the typical precipitation distribution in spring storms versus Cyprus lows; sharav cyclones; tropical systems affecting precipitation in the region; what are the typical precipitation intensities and how common is severe convection in such storms in this season; how common are cut-off lows?

Response: Following your comment, we plan to insert the following paragraphs in the introduction: The majority of the annual precipitation in Israel is associated with Mediterranean cyclones, while reaching its eastern part (i.e., Cyprus Lows, HMSO 1962; Saaroni et al. 2010; Zappa et al. 2015). Two-thirds of the rainfall occur during December through February (Alpert et al. 2004; Ziv et al. 2006). The focus on the Negev desert and Judean desert (Fig. 1), hereafter referred to as the 'study region'. The climatic regimes of the study region span from semiarid in the north to an arid in the center and the south (south of 31.25N, Ziv et al 2014). During the late spring (Apr-May), the rainfall over the northern and central parts of the Negev desert constitutes 4 - 9% of the annual average (5 - 10 mm). In spite of these negligible rain amounts, the number of flash flood events cannot be ignored. The flood regime in the study region

was analyzed by Kahana (1999), based on 37 hydrometric stations operated by the Israeli Hydrological Service. He identified 59 "major floods". A "major floods" is a flood in which the recorded peak discharge reached the magnitude of a 5-year recurrence interval for the period 1947- 1994 at least in one watershed. Eight (14%) of these major floods occurred during the late spring. The main synoptic circulation system associated with major floods in the late spring over the study region is the "Active Red Sea Trough" (ARST, Kahana et al., 2002). The ARST is most frequent during fall and spring (Sharon, 1978; Sharon and Kutiel, 1986; Dayan et al., 2001). This is a lower-level trough extending into the eastern and southern Israel from equatorial eastern Africa along the Red Sea and is accompanied by a pronounced quasi-stationary upper-level trough that develops over Egypt. At times, it initiates severe convective storms. The secondary source of major floods is generated mainly by a derivative of the Cyprus Lows — the Syrian Lows. These Lows are Mediterranean midlatitude cyclones that approach Syria and differ substantially from normal conditions, in which Mediterranean cyclones tend to decay before reaching Syria (Kahana et al, 2002). Under such atmospheric conditions, the surface north-westerly flow over the eastern Mediterranean is enriched with moisture and crosses the northern Negev perpendicular to the terrain upslope toward the Dead Sea. This enhances the generation of orographic convective rain over the southern Dead Sea basin and the Judean Desert. As for the Sharav cyclone, the only rain that induced major flood in the study area resulting from this synoptic system was documented by Kahana et al. (2002), in April 1971. Considering the relation between rain intensities and the dominating synoptic systems or the timing along the rain season, such a database does not exist yet. Kahana et al. (2002) distinguished between the major flood intensity associated with ARSTs and Syrian lows and found these associated with the ARST as being more intense. This suggests that the rain associated with the ARST is more intense. This notion will be included in the introduction and discussed against the extreme rain intensities observed in the present storm, in spite of its being associated with a Syrian low. Concerning cutoff lows, we intend to include the following paragraph in the introduction of the revised manuscript:

"The storm was associated with a Syrian low, accompanied by an upper-level closed cyclone, resembling the features of a 'cutoff low'. A cutoff low is defined as a closed cyclone in the upper levels, overlapping with a PV maximum (e.g., Hoskins et al. 1985). Cutoff lows are considered as favorable for severe weather over the Mediterranean Basin (Porcu et al, 2007). For instance, it was found as a major source for heavy rainfall on a daily basis in south Portugal (Fragoso and Tildes Gomes, 2008). Porcu et al. (2007) found a total of 273 cutoff lows during the period 1992–2001 over the entire Mediterranean Basin. Over the 30-35N, 30-35E domain, representing the Levant, they found 6 cases (average of 0.6 y-1) during the months April-September. Since the Levant is free of upper-level significant cyclonic system (and also of rain) during the summer months June-September (Kushnir et al. 2017). This result reflects, actually, the events of April – May." The question "how common are cut-off lows?" is addressed by a comparative analysis of 11 rainstorms that caused floods in the late spring over the study region (including the present one). We found that in 10 of them a cutoff low was involved. We plan to include the findings in a separate subsection in section 3 (results).

2. The paragraph describing the aim of the study (L46-48) should be clarified. It is currently not clear what the authors mean by "one of the latest spring severe events. . ." does "latest" refer to the most recent one? Or to severe precipitation occurring very late in the rain season? Especially when the introduction is sufficiently expanded, it should be more clearly outlined what is unique about this storm/flood. For example, how well was it forecasted? What is unusual about the distribution of precipitation? What is unusual about this cut-off low? What else is unusual beyond its fatal impact? What do the authors mean by "its unique features" (L47)? The authors should avoid using such general terms and be more specific.

Response: In the revised text we will refrain from giving superlatives to the storm. Its features in the climatological perspective will be given in the introduction. The part of the original text, at L46: "The aim of this study …. of the Mediterranean" will be

changed to: "This storm was severe in several aspects. One is the number of fatalities, 13, which is record breaking for Israel. Second, a part of the Zin basin (draining the center of the Negev desert) experienced precipitation intensities reaching 75-100-year return period, resulting in discharge magnitudes of 10-50-year return period (Rinat et al. 2020). The third aspect is the rainfall totals for the storm, which reached 40-50 mm over wide parts of the study region, i.e., 10 times the monthly long-term mean (retrieved from https://ims.gov.il/en/node/46). The aim of this study was not to evaluate the rainfall forecast's ability to predict this event (covered by Rinat et al. 2020) but rather to assess the severity of this storm in the seasonal perspective and to analyze the atmospheric conditions that explain its severity ". As for the role of cutoff lows in the climate of the Mediterranean, we plan to add a paragraph, quoted in our response to comment 22.

Reference: Rinat, Y., Marra, F., Armon, M., Metzger, A., Levi, Y., Khain, P., Vadislavsky, E., Rosensaft, M. and Morin E. Hydrometeorological analysis and forecasting of a 3-day flash-flood triggering desert rainstorm, NHESS doi.org/10.5194/nhess-2020-189, 2020.

3. The construction of the reference list of cases is not outlined with sufficient details. Are there objective quantitative criteria? Lines 58-59 and L 66-67 are still too general. Which streams are considered? What is the threshold for discharge? In how many stations? What is the reference region? Is the list restricted to cut-off lows? It will be good to reference Table 1 at this stage and provide information on the precipitation in those reference storms, to then contrast the current storm in focus that produces a very different precipitation distribution.

Response: The present paper has no hydrological orientation. This subject is covered in depth by Rinat et al. (2020). The storms referred to in line 58 are the same as these mentioned in line 66. To avoid ambiguity, the phrase "to other 11 storms, spread over 28 days, in which discharge was observed over the study area, during April and May for the years 1986 – 2018, entitled hereafter the 'reference period'" will be changed

to: "to a set of 11 storms (including the one studied here), spread over 30 days, in which discharge was observed over at least one of the 37 hydrometric stations of the Israeli Hydrological Service that has been operative during 1986 – 2018 over the study area. These storms, which occurred during April and May, are hereafter entitled as the 'reference storms'". These storms were further analyzed and the findings will be included in a subsection in the "results" section.

4. There is a somewhat inconsistent usage of reanalysis datasets, with some fields taken from NCEP/NCAR, but PV and omega taken from ERA5. Why not analyse all fields from ERA5?

Response: The reason for using different data sources is availability. Unfortunately, we have no access to ECMWF data of 2.5×2.5 deg resolution on the one hand, and there is no fine resolution NCEP data, in the order of 0.25×0.25 deg, on the other. However, a comparison of several parallel maps from the two data bases used shows a satisfying fit (see Fig. 1 below - example of SLP). We will note this in the revised manuscript.

5. The motivation for examining and comparing the MCV values is not clearly revealed. Why not consider the shear vorticity as well and take the commonly used relative vorticity as a measure for the intensity? Please justify this, especially given the fact that relative vorticity is anyway shown in Fig. 3. This clarification is again needed with regard to the list of reference cases. What do we learn from the high MCV? How do you interpret these differences?

Response: The 'measure of the curvature vorticity' (MCV) is a proxy for average curvature vorticity over the upper-level cyclone, so it evaluates its overall intensity. Since this cyclone moved along the latitudes of Israel, while the Subtropical jet shifted >7 deg southward, the jet's associated wind shear vorticity was regarded as marginal, so that the MCV can be considered as an estimate for the full vorticity (See Fig. 2 below). We will add this notion in the "methodology" section before introducing the MCV.

Minor comments

1. L13: "one of the latest. . .3 decades" this is not clear and appears again throughout the manuscript. Please rephrase and clarify if you refer to the late timing in the season or to longer time scales.

Response: The phrase "The timing of the storm is also unique, at the end of the rainy season, when rain is relatively rare and spotty. The study analyses the dynamic and thermodynamic conditions that made this rainstorm one of the latest spring severe events in the region during the last 3 decades" will be changed to: "The timing of the storm, at the end of the rainy season, when rain is relatively rare and spotty, raises the question what made this rainstorm so intense." The subject will be further treated in the revised introduction, see our response to major comment #2.

2. L25: delete the mention of the temperature anomaly which is not shown, or add a section with this evidence to the results.

Response: We opt for the second alternative, and therefore will leave the notion in the abstract as is and add to the revised manuscript the following: "In the case studied here, the instability can be attributed to a negative temperature anomaly in the upper levels (in the order of -5 K in 500 hPa, not shown) that covered the southern Levant, as a part of the cutoff low".

3. L43: what does "them" refer to?

Response: We will replace the word "them" by "the resulting rains".

4. L53: I suggest to replace "Material" by "Data and Methods"

Response: Will be replaced.

5. L57: delete "to"

Response: Will be done.

6. L58: what is meant by "maximum intensity"? it should be more accurate and clarify if it refers to precipitation/discharge/a vorticity measure/cyclone characteristic etc.

Response: After further consultation, we came to the conclusion that from the mete-orological point of view, no distinct difference can be noted in the intensity of the rain between the 25 and the 26 of April. The relevant notions will be removed from the entire manuscript. We will replace "when it reached its maximum intensity" in L58 by "when the Tzafit flood took place" as well.

7. Fig. 1 caption: indicate the times in UTC for the date range; add "the red start marks the location of . . ."

Response: Will be done.

8. Equations: replace the cross sign with a dot, to not confuse with vector notation.

Response: Will be done.

9. L83: how is the depth of the cyclone estimated? Please provide the accurate measure.

Response: The cyclones' depth was derived from the 500 hPa GPH. In the text, be-fore "The tangential. . ." we will insert "The cyclone depth (in meters) is the difference between the central height and that of the outer most isohypse (which are depicted in 15 m intervals)."

10. L92: add "temperature" after "mean".

Response: Will be done.

11. L94: remove one S from SSI.

Response: Will be done.

12. L89-103: for each index, mention which values indicate severe convection or thun-derstorms.

Response: In Fig.3 and in its caption, the terms "High", "Elevated" and "Low" were used to describe values highlighted in red, yellow and green, respectively. The grades are

based on values corresponding to severe convection for the pertinent region. For CAPE – 1000 j/Kg (Dayan et al. 2001), SI - -4C (Dayan and Sharon, 1980), LI – negative (Galway 1956). These values and references will be added to the figure caption. Ref: Dayan U., B. Ziv, A. Margalit, E. Morin, D. Sharon, 2001: A severe autumn storm over the middle-east: synoptic and meso-scale convection analysis, Theor. App. Clim., 69, 1/2, 103-122.

13. Eq 4: What is meant by RH850,700? why a modification of the K-index is needed for the eastern Mediterranean? Ziv, B., Harats, N., Morin, E. et al. Can severe rain events over the Mediterranean region be detected through simple numerical indices? Nat Hazards 83, 1197–1212 (2016). https://doi.org/10.1007/s11069-016-2385-y.

Response: RH850,700 is the average relative humidity of the 850- and 700-hPa levels. The MKI was proposed and examined by Harats et al. (2010), to reduce false thunderstorm alarms in the Mediterranean region. The MKI differs from the KI in that the 1st term, the temperature difference between 850 and 500 hPa, is multiplied by the average relative humidity over the 850 and 700 levels, so that when the lower- or med-levels are dry the MKI is reduced compared to the KI. The MKI was further elaborated and tested by Ziv et al. (2016). See below Fig. 3 demonstrate high KI values compared to MKI over arid regions for selected times of the storm (07 and 09 UTC).

14. L104: replace "also used" by "analysed". Add "as" before "if". Is PW based on ERA5?

Response: The PW was calculated from the soundings of Beit Dagan. We will state it in the revised manuscript.

15. L116: "which activated convection" – this statement is not backed by evidence at this stage and should be deleted.

Response: We agree and will omit this phrase.

16. L118-119: the term "precipitative elements" is not a clear.

Response: We meant "major rain cells". The text will be corrected accordingly along the entire text.

17. L121-122: the statement is again not backed by evidence at this stage, and it is not clear how this conclusion is reached, especially since it appears in the beginning of the results section. Furthermore, here and in L184-188 and throughout the text, the relationship between dynamical factors / orographic effects / convection / thermodynamic factors should be more clearly defined and distinguished from one another. For example, omega in ERA5 incorporates mass fluxes from convection, so its attribution as a clear dynamical diagnostic is not accurate. Please readdress these definitions, and outline them with regard to the analysis you carry out in this work.

Response: The sentence in L121-L122 is indeed a discussion statement, so it will be removed from the results (#3) section. Following your comments, we decided to refrain from using the Omega maps as a basis for explaining the formation of the major rain cells (hence, we omitted the Omega maps from Fig. 7). We propose a modified version in 2 aspects: 1) only two times (09 and 12 UTC) will be considered, corresponding to the two major rain cells analyzed. 2) the Omega maps are replaced by relative vorticity maps and satellite IR images, see Fig. 4 below. We are also reformulating the inter-relations among the various factors that controlled the convective activity in this rainstorm. This will be reflected in the discussion section.

18. L130-131: the transport of the dry air over the Levant is not consistent with the evolution of enhanced clouds at this stage.

Response: We do not agree on this point. As stated in the manuscript, prior to the initiation of the storm, on April 24, 2018, 12 UTC the Red Sea Trough (RST) at the lower-levels (Fig. 2a) created south-easterly flow across the lower and mid-levels, transporting dry air from the Arabian Desert toward the EM. This is evidenced by the Beit-Dagan radiosonde for that time indicating 30-45% relative humidity between 500-3,000 m ASL and below 10% between 3,000 and 5,000 m, and is consistent with the

absence of cloudiness in figure 2a. 12 hours later, on April 25, 2018 00 UTC, the relative humidity from the surface up to 5,000 m exceeded 50%, and in several layers even 70%. The resulting cloudiness in April 25, 12 UTC is seen in the satellite imagery (Fig. 2b).

19. Fig. 2: switch the locations of panels c and d; add "blue contours" after "m", and "black contours" after "hPa".

Response: Will be done.

20. Fig. 3: replace "Course" by "Track"; add initials to the caption, e.g. "precipitable water (PW), CAPE (CA)". . . ; replace here and throughout the manuscript (e.g., L200) "Km" with "km"; the blue text over the Med Sea is not visible; arrows in the late stages of the track are not visible;

Response: Following your suggestion, the term "course" will be replaced by "track" in the caption of Fig. 3, and initials will be added for the precipitable water and the 3 stability indices. Arrows were not marked in the late stages of the track due to the lack of space and for the sake of clarity. "Km" will be corrected to "km" here and along the entire text. The intended corrected caption is: "Figure 3: Track of the upper-level cyclone (500 hPa GPH) during 24–27 April 2018, in 12 hours increments. The instantaneous speed (ms-1) and distance (km) spanned during each increment are denoted by s and d, respectively. For each 24 hours increment, the radius of the upper-level low (km), the maximum relative vorticity (s-1), the precipitable water (PW, in mm) and 3 thermodynamic indices (LI, SI, CA) as calculated from the sounding of Beit Dagan are specified. "High", "Elevated" and "Low" values of the indices are highlighted in red, yellow and green, respectively."

21. Fig. 4 and accompanying text: it is unclear if this is PV or its anomaly (and how the anomaly is defined). I also recommend to switch the units to PVU and enlarge the domain. Response: The term "anomaly" refers to its spatial aspect (following Hoskins et al. 1985). This will be clarified in the text (see in #22). The units in this figure (see

Fig. 5 below) have been changed to PV-units and the domain enlarged.

22. L141-142: please add a reference to a climatology of such cutoff lows to demonstrate it is exceptional.

Response: We modified the relevant phrasing and prepared for the introduction the following background and climatological aspects of Cutoff lows: "A cutoff low is defined as a closed cyclone in the upper-levels, overlapping with a PV maximum (e.g., Hoskins et al. 1985). Cutoff lows are considered as favorable for severe weather over the Mediterranean Basin (Porcu et al, 2007). For instance, it was found as a major source for heavy rainfall on a daily basis in south Portugal (Fragoso and Tildes Gomes, 2008). Porcu et al. (2007) found a total of 273 cutoff lows during the period 1992–2001 over the entire Mediterranean Basin. Over the 30-35N, 30-35E domain, representing the Levant, they found 6 cases (average of 0.6 y-1) during the months April-September. Since the Levant is free of upper-level significant cyclonic system (and also of rain) during the summer months June-September (Kushnir et al. 2017). This result reflects, actually, the events of April – May." Unfortunately, there are no previous studies on the role of cutoff lows in the EM during the spring season.

Refs: Fragoso M, Tildes Gomes P. 2008. Classification of daily abundant rainfall patterns and associated large‐scale atmospheric circulation types in Southern Portugal. Int. J. Climatol. 28(4): 537–544, doi: 10.1002/joc.1564. Kushnir, Y., U. Dayan, B. Ziv, E. Morin, and Y. Enzel, 2017. "Climate of the Levant: phenomena and mechanisms" in "Quaternary of the Levant: Environments, Climate Change, and Humans", Y. Enzel and O. Bar-Yosef Editors. Cambridge University Press, London (pp 31-44). Porcu et al. (2007): A study on cutoff low vertical structure and precipitation in the Mediterranean region, Meteorol Atmos Phys 96, 121–140 (2007) DOI 10.1007/s00703-006-0224-5.

23. L165-166: This sentence is not clear. Can simplify by rewording to ". . . is expressed by enhanced easterly flow between the two vortices."

Response: We will adopt the rewording as suggested.

24. Fig. 5: In my view, the figure belongs more naturally in the discussion, and clearly after Fig. 6. In the figure, the term "blocking L" is confusing and should be reworded to "cut-off L".

Response: We replaced figures 5 and 6 by one figure (see below) and moved it to the discussion. We agree that the term "cutoff low" is more appropriate for the closed cyclone than "blocking Low". The figure and the text will be modified accordingly (See Fig. 6 below).

PV distribution on 300 hPa, starting from April 25 2018, 12 UTC, in 6 hour intervals, covering Europe and the Mediterranean Basin, on which a conceptual diagram of a dipole type block (following Yamazaki and Itoh, 2013) is superposed. A cutoff low is seen over the south-eastern Mediterranean and a blocking high over Eastern Europe, forming a dipole. The arrows show the induced flow of each of the two vortices.

25. In Fig. 6: the arrows in (c) are not visible.

Response: We intend to remove this figure (see our response to the previous comment #24).

26. Fig. 7 caption: add in the end ", of 26 April 2018"; add units of MKI.

Response: The units are deg C. The text will be modified accordingly.

27. L 189: "cloud systems rotated cyclonically" – is there evidence for this advection as opposed to locally-produced clouds?

Response: Following your reservation considering our proposed cyclonic rotation of the cloud system, we went through hourly radar and satellite images, and came to the conclusion that in spite of the cyclonic trajectory of the air that entered Tzafit in the height of 1,400 m (Fig. 9), there is no robust evidence for such a cloud cyclonic rotation. Hence, this hypothesis will be omitted. The only relevant reference for that is moisture

that originated from sources east of Israel that have contributed to the intensity of the rain in Tzafit, as reflected by the above mentioned air back-trajectory.

28. L187, 194: replace "ascendance" by "ascent".

Response: To be done.

29. L201: change 3c to 2c.

Response: To be done.

30. L202: add "reference" before "days"

Response: To be done.

31. Table 1: unclear how the depth is defined.

Response: The cyclones' depth was derived from the 500 hPa GPH and are, therefore, expressed in m. In the text, before "The tangential..." we will insert "The cyclone depth (in meters) is the difference between the central height and that of the outer most isohypse."

32. Section 3.3 and elsewhere: change "Kg" with "kg"

Response: To be done.

33. L226-227 "where it interacted with deep moist convection" this is a vague statement. Please clarify what you mean here.

Response: The sentence: "... moisture strip that extended from tropical latitudes to Saudi Arabia, ... in the morning of April 26 (Fig. 7e)." will be changed to: "The major feature is a moisture strip that extended from Saudi Arabia, curved cyclonically through Iraq, and entered north Israel from the east in the morning of April 26 (Fig. 8). This moisture transport can be inferred from the trajectory analysis (see Fig, 9)."

34. L233: what is the evidence for "One is of tropical. . . at upper levels"?

Response: After elaborating the possibility of moisture contribution from tropical sources, we decided to omit this hypothesis. The text will be modified accordingly.

35. L247: replace "one of the latest spring severe events" with "a severe storm occurring latest into spring".

Response: To be done.

36. L257-258: cutoff lows are not typical midlatitude cyclones, but rather a particular case in which the high-PV air is separated horizontally from the stratospheric reservoir in the upper troposphere.

Response: We omitted the notion of mid-latitudes in this sentence.

37. L268-269: Please comment on the timing. This is occurring one day before the flood.

Response: We will change the phrase " The instability indices reached values indicating potential for thunderstorms (CAPE = 909 J Kg-1, LI = - 4.9 K, SI = - 2.7 K and MKI = 30 K). At the same time, the precipitable water over the study area increased from 17 to 30 mm" to: "During 25-26 April, the instability and moisture indices reached values that reflect potential for thunderstorms, and varied between these two days as follows: CAPE= 778 to 909 J Kg-1, LI = -4.9 to -3.6 K, SI = -2.7 to +1.7 K and the precipitable water (PW) 29 to 26 mm".

38. L299: "-5 K temperature anomaly". Please add more details on where is this anomaly located, at what vertical level, and add "not shown".

Response: We will modify the phrase "a -5 K temperature anomaly over the region" to "a -4 K temperature anomaly in the 500 hPa level over the Levant (not shown)". See Fig. 7 below a map showing the temperature anomaly at 26 April 2018, 09 UTC.

[Figure]

[Figure]

**Fig. 1.** example of SLP: NCEP vs. ECMWF.

[Figure]

**Fig. 2.** The zonal component of the wind averaged over 25-27 April 2018 in 500 (left) and 300 (right) hPa

**MKI**

**KI**

**Fig. 3.** MKI vs. KI

**Fig. 4.** Set of maps for April 26 2018, 09 and 12 UTC: Satellite image of MSG ch9 (10.8 um) (a,b); Radar imagery of one hour integrated rain depth (mm), (c,d); Relative vorticity and wind (e,f) and MKI (g,h).

PVU ($10^{-6}$ K m$^{-2}$ kg$^{-1}$ s$^{-1}$)

**Fig. 5.** PVU and enlarged domain of org Fig. 4.

[Figure]

**Fig. 6.** PV distribution and conceptual diagram of a dipole type block

**Fig. 7.** temperature anomaly at 26 April 2018, 09 UTC.

---

## Referee Report (RR1)

General comments: The manuscript "Atmospheric Conditions Leading to an Exceptional Fatal Flash Flood in the Negev Desert, Israel" by Uri Dayan et al. deals with a high-impact severe weather situation across Israel. During the slow passage of an unseasonably intense cut-off trough, a three-day active weather period was observed what is not typical for this time of the year. Rain was produced by both stratiform and convective processes, with the convective storms in the southern portions yielding the highest impact. The authors describe the large-scale situation and present backward trajectories that indicate the advection of moist air masses towards Israel.

The manuscript improved significantly. However, there are still a few aspects that can be elaborated further.

First, the analysis and discussion on the development of instability can be more focused. From the manuscript, the main focus seems to be unseasonable cold mid-level air what is also associated with the intensity of the upper cut-off trough. As strong diurnal heating took place prior to convection initiation, one can argue that this diabatic heating lead to steep lapse rates. In the next step, the overlap of these lapse rates with the rich moisture can be analysed. Finally, indices like CAPE, KI and MKI might be presented to highlight the area of instability.

Second, lift that causes the storms to initiate needs to be addressed. Mesoscale circulations need to be taken into account and it is important to discuss if baroclinic circulations (fronts, outflow boundaries) were present or not. Then, the influence of the topography can be addressed as already presented (lines 323-324). Are there indications that storms moved along such a boundary where low-level convergence is maximized?

Third, next to convection initiation, the potential for storms to produce high amounts of rain needs to be discussed. The amount of rain is dependent on rain intensity (that is also dependent of rain efficiency) and rain duration. From the radar images, it looks like the individual storms were not too large. Given the very localized rain maxima, it may be possible that the slow storm motion contributed to the high rain accumulations. The slow storm motion might be connected to the position of the cut-off trough as the trough center was located over the area of severe storms. The rain efficiency can be also discussed, e.g. with respect to limited entrainment given a relatively moist profile (with a history of deep moist convection in the days before) and the vertical distribution of CAPE (i.e. skinny CAPE profiles that allow for slow ascent and much time for the rain production process). Moreover, the data can be used to discus the potential for a warm rain process within a deep warm cloud layer, e.g. due to the high temperature at low levels and the rich moisture content at both low- and mid-levels.

---

## Referee Report (RR2)

The publication has made great progress. In the current version, ingredients of deep moist convection are now discussed. Particularly promising is the fact that a high-resolution model can be used to analyse the floods. I therefore recommend another iteration of the review to include one more figure derived from the model.

1. Thermodynamic profile derived from the model

I recommend to include a representative vertical profile for this case, e.g. derived from the model. Are there characteristics of heavy rain events, like modest CAPE, skinny CAPE profiles (e.g. small negative values of LI), high relative humidity at mid levels, a deep warm cloud layer due to a low lifted condensation level and a freezing level high above the ground (e.g. 3 km), weak CIN, weak flow and expected slow storm motion? Recognition of the mentioned characteristics can improve forecasts of such events, together with the given large-scale analysis and the presented MKI maps.

2. Convection initiation

It would be good to discuss the reasons that may have led to training storms over the southern location as shown in figure 9 (left). For example, due to the northerly low-level winds, there could be upslope flow at the southern end of the valley. Additionally, the developing mountain-valley circulation short after sunrise might have supported new storm development parallel to the mountain slopes at the western flank of the valley that led to the flood later at 9:35 UTC. It would add to the discussion of large-scale forcing.

---

## Author Response (AR2)

Point to point response (in red)

First, we wish to thank the reviewer for urging us to elaborate on the factors that controlled and enhanced the rain centers, and for the valuable advices. Following your review, we used a regional model to further analyze mesoscale processes and inspected the radar and satellite imagery. As you will note, we have gained several new findings, which are specified below and added to the text in red. An additional paragraph lists the factors that explain the high rainfall observed in this storm. We also replaced several parts of Fig. 8 and added 2 new figures.

General comments: The manuscript "Atmospheric Conditions Leading to an Exceptional Fatal Flash Flood in the Negev Desert, Israel" by Uri Dayan et al. deals with a high-impact severe weather situation across Israel. During the slow passage of an unseasonably intense cut-off trough, a three day active weather period was observed what is not typical for this time of the year. Rain was produced by both stratiform and convective processes, with the convective storms in the southern portions yielding the highest impact.

The authors describe the large-scale situation and present backward trajectories that indicate the advection of moist air masses towards Israel. The manuscript improved significantly. However, there are still a few aspects that can be elaborated further.

Following your comments, we added to our analysis "the COSMO regional model (www.cosmo-model.org), operational at the IMS, covering the Eastern Mediterranean domain (25-39E/26-36N, Khain et al., 2020b) with 2.8×2.8 km resolution. The model analyses are created by using a continuous assimilation of the radar and rain gauges composite via latent heat nudging (Stephan et al., 2008, Khain et al., 2020a). The boundary and initial conditions are taken from the European Centre for Medium-Range Weather Forecasts (ECMWF) Integrated Forecasting System (IFS) (www.ecmwf.int/en/forecasts/documentation-and-support). The COSMO model is based on the primitive thermo-hydrodynamic equations that describe non-hydrostatic compressible flow in a moist atmosphere. Its vertical extension reaches 23.5 km (~30 hPa) with 60 model levels, including 12 levels between the surface and 900 hPa and 15 levels between 900 and 500 hPa, being able to capture the PBL effects (Uzan et al., 2020)."

The above description of the model is added as the 4th paragraph of section 2 in our revised manuscript. For more details, see:

http://www.cosmo-model.org/content/model/documentation/core/default.htm

We also further elaborated the radar and satellite imagery to gain the detailed evolution of the major rain centers.

First, the analysis and discussion on the development of instability can be more focused. From the manuscript, the main focus seems to be unseasonable cold mid-level air what is also associated with the intensity of the upper cut-off trough. As strong diurnal heating took place prior to convection initiation, one can argue that this diabatic heating lead to steep lapse rates. In the next step, the overlap of these lapse rates with the rich moisture can be analysed. Finally, indices like CAPE, KI and MKI might be presented to highlight the area of instability.

The warming during the day hours of 26 April 2018 is demonstrated by the 925 hPa temperature. The maps for 07, 09 and 12 UTC (corresponding to 09, 11 and 14 LST) show a

gradual warming during the hours when the sky was clear. The hours 09 and 12 UTC correspond to the beginning of the rain events in Tzafit and Beit Shean, respectively.

[Figure]

Temperature at 925 hPa for 26 Apr 2018 07, 09 and 12 UTC (left to right). Tzafit is denoted by a white circle and Beit Shean – by a red one.

The spatial distribution of muCAPE (most unstable CAPE) was derived from the COSMO model output, since the only sounding station in Israel is that of Beit Dagan, only twice a day (as is mentioned in the manuscript). The muCAPE shows a general increase during the day hours inland, especially between 10 and 12 UTC (12 and 14 LST). A pronounced maximum, of 1200 J/kg with an elongated shape, is seen north of Tzafit, along the path of the rain producing cells in 09 UTC (north-south, see below), just before the rain started there. This maximum moved eastward, weakened and changed structure later. As for Beit Shean, a buildup of a pronounced muCAPE maximum, of >1500 J/kg, is seen around it in 12 UTC, just before the rain started there.

[Figure]

Spatial distribution of muCAPE for 26 Apr 2018 07, 09 and 12 UTC (left to right). Tzafit is denoted by a white circle and Beit Shean by a red one.

The thermodynamic profile derived from the COSMO model at the location of Tzafit for 09 UTC was plotted, and the CAPE and LI were calculated manually, yielding 1075 J/kg and 4.0 K respectively. The respective values for Beit Shean at 12 UTC (prior to the rain there) were 1740 J/kg, and 4.5K, respectively.

The daily warming as a factor for the building up of instability is now mentioned in the abstract and is included in discussion section 4, (lines 368-372), as a part of the 1st in the list of factors that explain the high observed rainfall, as follows: " The considerable rainfall in the three major rain centers described above can be explained by several factors: The high degree of instability, originated by the upper-level cold air, which was further enhanced by the intense solar radiation prior to the three rain events, as implied by the date, latitude and the clear sky (Fig. 8a). This effect is reflected by the MKI maps (Figs. 8g,h), showing much higher values inland compared to these along the coastal region (35°C in Tzafit Basin compared to 13.2°C in Beit Dagan). This is also reflected by a gradual increase in CAPE during the morning hours (not shown), up to 1740 Jkg$^{-1}$ in Beit Shean."

The maximal CAPE value is addressed first in the abstract. CAPE values are also emphasized in Sec. 3.1 (line 238) and in Sec. 3.2 (line 300).

The MKI map for 09 UTC, below, shows high values (>35K) north of Tzafit, where the flood producing rain clouds were formed. The MKI over Judean Mts. and Beit Shean at that time was marginal for intense convection. At 12 UTC, however, just before the rains began in Beit Shean and Judean Mts., the MKI in the 3 locations reached 35°C, i.e., favorable for intense convection.

[Figure]

MKI distribution for 26 Apr 2018 09 and 12 UTC. The white and red dots denote Tzafit and Beit Shean, respectively, and the green represents the Judean Mts.

These new MKI maps replace now the previous ones (based on the ERA5 with 30 km resolution data) included in Fig. 8. The values are mentioned in Sec. 3.2 (lines 300-301) and in the discussion on the instability as one in the list of factors explaining the high rainfall in section 4 (lines 369-372).

Following your request, below are the respective KI fields. The distribution is uniform as compared to that of the MKI. The low relative humidity in 700 hPa over the northwestern part of the study region (see RH map to the right) explains the relative low MKI compared to the KI there at 12 UTC. This demonstrates why KI gives false indications for intense convection when the mid-levels are dry (and cause entrainment of dry air into the forming clouds).

[Figure]

KI distribution for 26 Apr 2018 09 and 12 UTC and relative humidity at 700 hPa for 12 UTC (from left to right). The dots in the KI maps are as in the MKI maps.

Second, lift that causes the storms to initiate needs to be addressed. Mesoscale circulations need to be taken into account and it is important to discuss if baroclinic circulations (fronts, outflow boundaries) were present or not. Then, the influence of the topography can be addressed as already presented (lines 323-324). Are there indications that storms moved along such a boundary where low-level convergence is maximized?

As for mesoscale lifting mechanism for the storm initiation, the vorticity field shown in Fig. 8 of the manuscript (inserted below) reflects two features, which may explain the rains in Tzafit and Beit Shean. The rain in Tzafit can be attributed to a positive vorticity anomaly over the Dead Sea (denoted in blue, north-east of Tzafit), with ~50 km diameter, east of Tzafit (the white point) in 09 UTC. This implies positive vorticity advection via the northeasterly flow at that level. The rain in Beit Shean can be attributed to a pronounced vorticity positive anomaly over the north-west part of Jordan that approached that region (the red point) in 12 UTC from the east.

[Figure]

Vorticity and streamlines in 500 hPa, taken from fig. 8 in the manuscript.

The existence of positive vorticity advection is addressed first in section 3.2 in association with the rain in Tzafit in lines 273-275 and in Beit Shean in lines 290-291. This effect is also described in section 4, (lines 335-339) as follows: "Each of these two centres was found to be associated with a positive vorticity advection ahead of a positive anomaly in the 500 hPa level in the ERA5 maps with 30×30 km resolution. The first anomaly was located east of Tzafit at 09 UTC, prior to the rain there (Fig 8e), and the other, east of Beit Shean at 12 UTC, when the rain system started its movement southward over that region (Fig. 8f)."

Concerning the lower levels, for Tzafit, the 925-hPa wind field for 09 UTC, according to the COSMO model (the 1st figure of this document), does not show any specific feature that can indicate a lifting mechanism for the rain there, or north of it. As for Beit Shean, during the rain episode, a lower-level trough extended toward the region from the southeast (Jordan). Its curved shape and orientation (denoted by a blue line) resemble these of the cloud line that caused the rain. This trough and the cloud system that crossed the region southward during that time are shown below, and added to the manuscript, together with a representative radar image, as Fig. 10.

[Figure]

Successive radar images (in mm/h units), in a 20-min intervals, indicating the progression of the precipitative element that crossed Beit Shean (Lake Kinneret is seen northeast of it). Note that the maximum rain intensities reach 50 mm/h. To the right, 850-hPa streamline representing the wind field, with a blue curved line denoting the trough line. Beit Shean is denoted by a white dot in the wind map and in the radar image of 12:20 UTC.

This system is described now in section 3.2 (lines 287-289) as follows: "During this rain episode, a lower-level trough extended toward the region from the southeast (Jordan, Fig. 10c). Its curved shape and orientation (denoted by a blue line) resemble these of the cloud line". It is also mentioned in section 4, in lines 339-341.

Third, next to convection initiation, the potential for storms to produce high amounts of rain needs to be discussed. The amount of rain is dependent on rain intensity (that is also dependent of rain efficiency) and rain duration. From the radar images, it looks like the individual storms were not too large. Given the very localized rain maxima, it may be possible that the slow storm motion contributed to the high rain accumulations. The slow storm motion might be connected to the position of the cut-off trough as the trough center was located over the area of severe storms.

The rain in Tzafit was contributed by several successive cells, 15×15 km on the average, that crossed the region from north to south at an average speed of 12 ms$^{-1}$ (Rinat et al. 2020). This scenario is demonstrated by a series of radar imageries, showing the second cell. The total rainfall for this episode, based on radar echoes, due to lack of rain gauges at the specific, is shown below, having a maximum of 50 mm. These high values can be attributed to the cell propagation vector (train effect), i.e., repeated areas of rain that move over the same region in a relatively short period of time, which may cause flash flooding (e.g., Cordifi et al., 1996).

[Figure]

Successive radar images, in a 5-min increments, indicating the progression of a precipitative element crossing the Zafit basin (1 pixel is equivalent to 0.25 km$^2$). The time notations (5 min interval) refer to the summer clock, UTC+3h (with the curtesy of Yair Rinat). The units are mm/h, based on the conversion method of Mara and Morin (2015).

[Figure]

Total rain amount accumulated between 11:40 and 13:40 LST. Note the meridional orientation of the radar images pointing at the "training effect" (with the curtesy of Yair Rinat).

A combination of the two figures above is now added to the manuscript, as Fig. 9. The description of the "train effect" appears in section 3.2, lines 271-273, as follows: ".. evolution is represented by a series of radar images (Fig. 9), showing that it resulted from the passage of several cells that crossed the region from north to south, with an average speed of 12 ms$^{-1}$." It is also included in section 4 (lines 378-380) as factor (e) in explaining the high rainfall in this storm.

As for Beit Shean, here the propagation speed of the rain system was indeed slow (~7 ms$^{-1}$), as can be inferred from the sequence of radar images (above). The rainfall was in the order of 50 mm, with a maximum of 72 mm, presumably contributed by both, the slow movement of the cloud system and the extreme rain intensity (up to 20 mm/h for a 10 min average, Porat et al. 2018). This is noted now in section 3.2, (lines 286-290) and in section 4, (lines 381-382) as the last one in the list of factors explaining the high rainfall in the storm: "The high rainfall in Beit Shean can be partly explained by the slow movement of the rain system, at a speed of 7 ms$^{-1}$."

The rain efficiency can be also discussed, e.g. with respect to limited entrainment given a relatively moist profile (with a history of deep moist convection in the days before) and the vertical distribution of CAPE (i.e. skinny CAPE profiles that allow for slow ascent and much time for the rain production process).

The vertical profiles plotted manually for both Tzafit and Beit Shean indicate that the temperature of the raised air parcels was considerably higher than its surrounding within the medium levels, as reflected by the LI (both exceeding 4K). This implies that slow ascendance was not involved. This and the high CAPE values (>1000 J/kg) indicate that the convection was vigorous.

Moreover, the data can be used to discus the potential for a warm rain process within a deep warm cloud layer, e.g. due to the high temperature at low levels and the rich moisture content at both low- and mid-levels.

[Figure]

Satellite thermal image for 26 April 2018 (based on METEOSAT IR channel) for the time when the rain attained its maximum intensity (left to right) in Tzafit and Beit Shean (shown by

arrows). The rain clouds over Judean Mts. are seen in the image of 1230 UTC south-west of Beit Shean.

The satellite IR images for the time of the rain in Tzafit and Beit Shean indicate that the cloud tops temperature reached -50°C. The vertical profiles indicate that their elevation exceeded 9000 m, which is 2500 m higher than the average cloud tops characterizing winter thunderstorms in Israel (Altaratz et al. 2001). Since the freezing level during this event was slightly below 3000 m, it can be concluded that the majority of the cloud cells contained a considerable amount of ice particles so that a warm rain process does not seem to be one of the factors for the high intensities observed.

The above METEOSAT satellite images replace now the previous ones in Fig. 8 (a,b). The low top temperatures are mentioned now in section 3.2, (lines 302-304): "… the tops of clouds of the three precipitative centers reached -50°C. According to the temperature profiles over the study region during the storm, this implies that these clouds exceeded 9000 m elevation." This factor is the second in the list of factors that explain the high observed rainfall in section 4 (lines 373-374): "The high elevation of the cloud tops in the three major rain centers, exceeding 9000 m ASL., 2500 m higher than that typifying the tops of the winter thunderclouds in Israel (Altaratz et al., 2001)."

**References**

Altaratz, O., Levin, Z., Yair, Y. 2001: Winter thunderstorms in Israel: A study with lightning location systems and weather radar, Mon. Wea. Rev., 129: 1259-1266.

Corfidi, S. F., J.H. Merritt, and J.M. Fritsch, 1996: Predicting the movement of mesoscale convective complexes. Weather and Forecasting, 11, 41-46.

Marra, F., and Morin, E.: Use of radar QPE for the derivation of Intensity–Duration frequency curves in a range of climatic regimes, Journal of Hydrology 531, 427–440. http://dx.doi.org/10.1016/j.jhydrol. 2015. 08.064, 2015.

Porat, A, Halfon, N and Forshpan A.: Examining the exceptionality of 25-27 April 2018 storm, IMS August (in Hebrew), 2018.

Rinat, Y., Marra, F., Armon, M., Metzger, A., Levy, Y., Khain, P., Vadislavsky, E., Rosensaft, M., and Morin, E., 2020: Hydrometeorological analysis and forecasting of a 3-day flash-flood-triggering desert rainstorm, https://doi.org/10.5194/nhess-2020-189, Preprint. Discussion started: 2 July 2020.

---

## Author Response (AR3)

**Point to point response** (in blue)

We wish to thank the reviewer for the valuable comments. Following your review, we add figure 11 and text as follows.

The publication has made great progress. In the current version, ingredients of deep moist convection are now discussed. Particularly promising is the fact that a high-resolution model can be used to analyse the floods. I therefore recommend another iteration of the review to include one more figure derived from the model.

**1. Thermodynamic profile derived from the model**

I recommend to include a representative vertical profile for this case, e.g. derived from the model. Are there characteristics of heavy rain events, like modest CAPE, skinny CAPE profiles (e.g. small negative values of LI), high relative humidity at mid levels, a deep warm cloud layer due to a low lifted condensation level and a freezing level high above the ground (e.g. 3 km), weak CIN, weak flow and expected slow storm motion? Recognition of the mentioned characteristics can improve forecasts of such events, together with the given large-scale analysis and the presented MKI maps.

**AU:** We added two vertical profiles based on the COSMO model output (new Fig. 11), one corresponds to the Tzafit flood and the other to the major rain center that hit Beit Shean. On each profile we drew a line denoting the adiabatic cooling of air parcel lifted from the surface. In both of them the CIN is negligible and the air parcel is considerably warmer than its surroundings throughout several kilometers. This reflects high CAPE values (>1000J/Kg) and implies high vertical velocity within the convective clouds. Moreover, as mentioned in the manuscript, the tops of clouds of the three precipitative centers reached -50°C. According to the temperature profiles over the study region during the storm, this implies that these cloud tops exceeded an elevation of 9000 m. This finding can explain the exceptional rain rates observed in the three major rain cells. These notions appear now in section 3.3 and mentioned in section 4. As for the weak flow, it is already addressed in the text through the wind map and the motion of the rain clouds.

[Figure]

**Fig. 11: Vertical profiles for (a) Tzafit (26 April 2018 0930 UTC) and (b) Beit Shean (26 April 2018 1230 UTC). The black lines denote temperature, the red dotted lines – dew point and the blue dashed lines show the adiabatic cooling of air parcels ascending from the surface.**

**2 .Convection initiation**

It would be good to discuss the reasons that may have led to training storms over the southern location as shown in figure 9 (left). For example, due to the northerly low-level winds, there could be upslope flow at the southern end of the valley. Additionally, the developing mountain-valley circulation short after sunrise might have supported new storm development parallel to the mountain slopes at the western flank of the valley that led to the flood later at 9:35 UTC. It would add to the discussion of large-scale forcing.

**AU:** The upslope effect of the Tzafit valley on the rain cells imbedded in northerly winds is questionable, because the diameters of the precipitative elements (seen in Fig. 9a) are in the same order of the valley width from north to south. It could be identified in the integrated rainfall map (Fig. 9b), showing two pixels (of 2.5×2.5 Km) with the same maximum values: one in the bottom of the valley and other on the southern slopes. This signal seems too weak to be noted.

As for the mountain-valley circulation, we added it to the list of optional explanations for the Tzafit rain event in Section 4 as follows: "An optional factor that may explain the repetitive formation of rain cells north of Tzafit is a mountain-valley circulation (anabatic) uplift over the eastern slopes of the ridge that extends from Judean mountains southward (Fig. 1). This could be expected in light of the clear skies prior to this event, but did not find any signature in the output of the COSMO model".